# Real-world Image Dehazing with Coherence-based Pseudo Labeling and Cooperative Unfolding Network

**Chengyu Fang**[1,*], **Chunming He**[1,3,*,†], **Fengyang Xiao**[1,2], **Yulun Zhang**[4,†],
**Longxiang Tang**[1], **Yuelin Zhang**[5], **Kai Li**[6], **Xiu Li**[1,†]

[1]Shenzhen International Graduate School, Tsinghua University, [2]Sun Yat-sen University,
[3]Duke University, [4]Shanghai Jiao Tong University,
[5]The Chinese University of Hong Kong, [6] Meta Reality Labs
✉ *chengyufang.thu@gmail.com*

## Abstract

Real-world Image Dehazing (RID) aims to alleviate haze-induced degradation in real-world settings. This task remains challenging due to the complexities in accurately modeling real haze distributions and the scarcity of paired real-world data. To address these challenges, we first introduce a cooperative unfolding network that jointly models atmospheric scattering and image scenes, effectively integrating physical knowledge into deep networks to restore haze-contaminated details. Additionally, we propose the first RID-oriented iterative mean-teacher framework, termed the Coherence-based Label Generator, to generate high-quality pseudo labels for network training. Specifically, we provide an optimal label pool to store the best pseudo-labels during network training, leveraging both global and local coherence to select high-quality candidates and assign weights to prioritize haze-free regions. We verify the effectiveness of our method, with experiments demonstrating that it achieves state-of-the-art performance on RID tasks. Code will be available at `https://github.com/cnyvfang/CORUN-Colabator`.

## 1 Introduction

Real-world image dehazing (RID) is a challenging task that aims to restore images affected by complex haze in real-world scenarios. The goal is to generate visual-appealing results while enhancing the performance of downstream tasks [1, 2]. The atmospheric scattering model (ASM), providing a physical framework for real-world dehazing, is formulated as follows:

$$P(x) = J(x)t(x) + A(1 - t(x)), \quad (1)$$

where $P(x)$ and $J(x)$ are the hazy image and the haze-free counterpart. $A$ signifies the global atmospheric light. $t(x)$ characterizes the transmission map reflecting varying degrees of haze visibility across different regions.

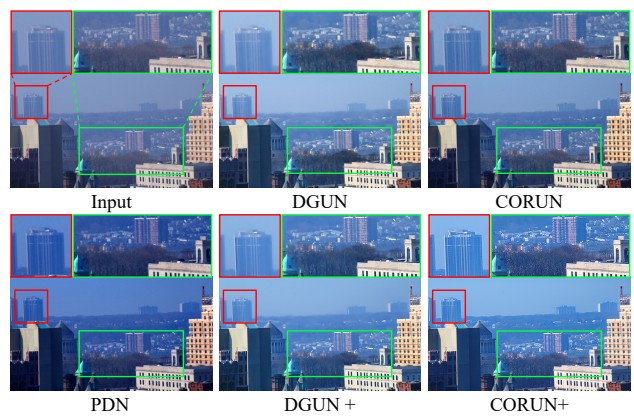

Figure 1: Results of cutting-edge methods. Our CORUN better restores hazy-contaminated details. Furthermore, techniques optimized by our Colabator framework, indicated by a "+" suffix, exhibit strong generalization in haze removal and color correction.

---

*Equal Contribution, † Corresponding Author, ✉ Email Address

38th Conference on Neural Information Processing Systems (NeurIPS 2024).

Conventional methods [3, 4] are limited by fixed feature extractors, which struggle to handle the complexities of real haze. Although existing deep learning-based methods [5–9] demonstrate improved performance, they face two significant challenges: (1) These methods do not accurately model the complex distribution of haze, leading to color distortion, as illustrated in fig. 1 DGUN [10]. (2) Real-world settings lack sufficient paired data for network training while optimizing the network with synthesized data brings a domain gap, limiting the generalizability of the models.

To overcome the first challenge, PDN [11] first introduces unfolding network [12, 13] to the RID field. In specific, PDN unfolds the iterative optimization steps of an ASM-based solution into a deep network for end-to-end training, incorporating physical information into the deep network. However, PDN does not effectively leverage the complementary information between the dehazed image and the transmission map, bringing overfitting problems and resulting in detail blurring (see fig. 1).

In this paper, we introduce the COopeRative Unfolding Network (CORUN), also derived from the ASM-based function, to address PDN's limitations and better model real hazy distribution. CORUN cooperatively models the atmospheric scattering and image scene by incorporating Transmission and Scene Gradient Descent Modules at each stage, corresponding to each iteration of the traditional optimization algorithm. To prevent overfitting, we introduce a global coherence loss, which constrains the entire pipeline to adhere to physical laws while alleviating constraints on the intermediate layers. These design choices collectively ensure that CORUN effectively integrates physical information into deep networks, thereby excelling in restoring haze-contaminated details, as depicted in fig. 1.

To enhance generalizability in real-world scenarios, we introduce the first RID-oriented iterative mean-teacher framework, named Coherence-based label generator (Colabator), designed to generate high-quality dehazed images as pseudo labels for training dehazing methods. Specifically, Colabator employs a teacher network, a dehazing network pretrained on synthesized datasets, to generate dehazed images on label-free real-world datasets. These restored images are stored in a dynamically updated label pool as pseudo labels for training the student network, which shares the same structure as the teacher network but with distinct weights. During network training, the teacher network generates multiple pseudo labels for a single real-world hazy image. We propose selecting the best labels to store in the label pool based on visual fidelity and dehazing performance.

To achieve this, we design a compound image quality assessment strategy tailored to the dehazing task, evaluating the global coherence of the dehazed images and selecting the most visually appealing ones without distortions for inclusion in the label pool. Additionally, we propose a patch-level certainty map to encourage the network to focus on well-restored regions of the dehazed pseudo labels, effectively constraining the local coherence between the outputs of the student model and the teacher model. As shown in fig. 1, Colabator, generating high-quality pseudo labels for network training, enhances the student dehazing network's capacity for haze removal and color correction.

Our contributions are summarized as follows:

(1) We propose a novel dehazing method, CORUN, to cooperatively model the atmospheric scattering and image scene, effectively integrating physical information into deep networks.

(2) We propose the first iterative mean-teacher framework, Colabator, to generate high-quality pseudo labels for network training, enhancing the network's generalization in haze removal.

(3) We evaluate our CORUN with the Colabator framework on real-world dehazing tasks. Abundant experiments demonstrate that our method achieves state-of-the-art performance.

## 2 Related Works

### 2.1 Real-world Image Dehazing

The dissonance between synthetic and real haze distributions often hinders existing Learning-based dehazing methods [14–18] from effectively dehazing real-world images. Consequently, there's a growing emphasis on tackling challenges specific to real-world dehazing [19–23].

Given the characteristics of real haze, RIDCP [7] and Wang *et al.* [24] proposed novel haze synthesis pipelines. However, relying solely on synthetic data limits models' robustness in real-world dehazing scenarios. Recognizing the distributional disparities between synthetic and real haze, methods like CDD-GAN [25], D4 [26], Shao *et al.* [27], and Li *et al.* [28] have utilized CycleGAN [29] for

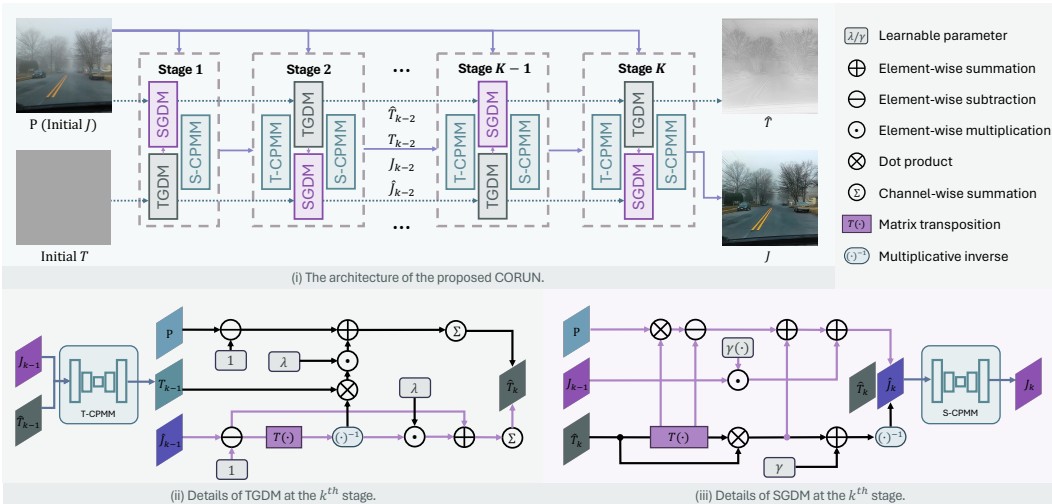

Figure 2: The architecture of the proposed CORUN with the details at $k^{th}$ stage.

dehazing. Despite this, the challenges inherent in GAN [30] training often result in artifacts. Some approaches combine synthetic and real-world data, applying unsupervised loss to supervise real-world dehazing learning [19]. However, these losses lack sufficient precision, leading to suboptimal results. Other methods leverage pseudo-labels [31, 32], but the erroneous pseudo-labels cause degrade quality.

To address these challenges, we introduce a coherence-based pseudo labeling method termed Colabator. Our approach selectively identifies and prioritizes high-quality regions within pseudo labels, leading to enhanced robustness and superior generation quality for real-world image dehazing.

## 2.2 Deep Unfolding Image Restoration

Deep Unfolding Networks (DUNs) integrate model-based and learning-based approaches [33, 34] and thus offer enhanced interpretability and flexibility compared to traditional learning-based methods. Increasingly, DUNs are being utilized for various image tasks, including image super-resolution [35], compressive sensing [36], and hyperspectral image reconstruction [37]. DGUN [10] proposes a general form of proximal gradient descent to learn degradation. However, it fails to decouple prior knowledge, relying solely on single-path DUN to model degradation and construct mappings, posing challenges in comprehending complex degradation. Yang and Sun first introduced DUNs to the image dehazing field and proposed PDN [11]. However, PDN does not exploit the complementary information between the dehazed image and the transmission map, resulting in detail blurring. Our CORUN optimizes the atmospheric scattering model and the image scene feature through dual proximal gradient descent, thus preventing overfitting and facilitating detail restoration.

## 3 Methodology

### 3.1 Cooperative Unfolding Network

We propose the Cooperative Unfolding Network (CORUN), the first Deep Unfolding Network (DUN) method utilizing Proximal Gradient Descent (PGD) to optimize image dehazing performance by leveraging the Atmospheric Scattering Model (ASM) and neural image reconstruction in a cooperative manner. Each stage of CORUN includes Transmission and Scene Gradient Descent Modules (T&SGDM) paired with Cooperative Proximal Mapping Modules (T&S-CPMM). These modules work together to model atmospheric scattering and image scene features, enabling the adaptive capture and restoration of global composite features within the scene.

According to eq. (1), given a hazy image $\mathbf{P} \in \mathbb{R}^{H \times W \times 3}$, we initialize a transmission map $\mathbf{T} \in \mathbb{R}^{H \times W \times 1}$. In gradient descent, we simplify the atmospheric light $A \in \mathbb{R}^3$ and implicitly estimate it in the CORUN pipeline to focus on the detailed characterization of the scene and the relationship between volumetric haze and scene. Hence, eq. (1) can be rewrite as

$$\mathbf{P} = \mathbf{J} \cdot \mathbf{T} + \mathbf{I} - \mathbf{T}, \tag{2}$$

Where $\mathbf{J}$ means the clear image without hazy, $\mathbf{I}$ is the all-one matrix. Based on eq. (2), we can define our cooperative dehazing energy function like

$$L(\mathbf{J}, \mathbf{T}) = \frac{1}{2}\|\mathbf{P} - \mathbf{J} \cdot \mathbf{T} + \mathbf{T} - \mathbf{I}\|_2^2 + \psi(\mathbf{J}) + \phi(\mathbf{T}), \tag{3}$$

where $\psi(\mathbf{J})$ and $\phi(\mathbf{T})$ are regularization terms on $\mathbf{T}$ and $\mathbf{J}$. We introduce two auxiliary variables $\hat{\mathbf{T}}$ and $\hat{\mathbf{J}}$ to approximate $\mathbf{T}$ and $\mathbf{J}$, respectively. This leads to the following minimization problem:

$$\{\hat{\mathbf{J}}, \hat{\mathbf{T}}\} = \underset{\mathbf{J}, \mathbf{T}}{\arg\min}\, L(\mathbf{J}, \mathbf{T}). \tag{4}$$

**Transmission optimization.** Give the estimated coarse transmission map $\mathbf{T}$ and dehazed image $\hat{\mathbf{J}}_{k-1}$ at iteration $k-1$, the variable $\mathbf{T}$ can be updated as:

$$\mathbf{T}_k = \underset{\mathbf{T}}{\arg\min}\frac{1}{2}\left\|\mathbf{P} - \hat{\mathbf{J}}_{k-1} \cdot \mathbf{T} + \mathbf{T} - \mathbf{I}\right\|_2^2 + \phi(\mathbf{T}). \tag{5}$$

We construct the proximal mapping between $\hat{\mathbf{T}}$ and $\mathbf{T}$ by a encoder-decoder like neural network which we named T-CPMM and denoted as $\mathrm{prox}_\phi$:

$$\mathbf{T}_k = \mathrm{prox}_\phi(\mathbf{J}_{k-1}, \hat{\mathbf{T}}_k), \tag{6}$$

the auxiliary variables $\hat{\mathbf{T}}$, which we calculate by our proposed TGDM can be formulated as:

$$\hat{\mathbf{T}}_k = \sum_{c \in \{R,G,B\}} (\mathbf{I} - \hat{\mathbf{J}}_{k-1}^c + \frac{\lambda_k}{(\mathbf{I} - \hat{\mathbf{J}}_{k-1}^c)^\top})^{-1} \cdot (\mathbf{I} - \mathbf{P}^c + \frac{\lambda_k \mathbf{T}_{k-1}}{(\mathbf{I} - \hat{\mathbf{J}}_{k-1}^c)^\top}). \tag{7}$$

The variable $\lambda_k$ is a learnable parameter, we enable CORUN to learn this parameter at each stage during the end-to-end learning process, allowing the network to adaptively control the updates in iteration.

**Scene optimization.** Give $\hat{\mathbf{T}}_k$ and $\mathbf{J}$, the variable $\mathbf{J}$ can be updated as:

$$\mathbf{J}_k = \underset{\mathbf{J}}{\arg\min}\frac{1}{2}\|\mathbf{P} - \mathbf{J} \cdot \hat{\mathbf{T}}_k + \hat{\mathbf{T}}_k - \mathbf{I}\|_2^2 + \psi(\mathbf{J}). \tag{8}$$

Same as the proximal mapping process in the transmission optimization, S-CPMM has the similar structure as T-CPMM but different inputs, we denote S-CPMM as $\mathrm{prox}_\psi$:

$$\mathbf{J}_k = \mathrm{prox}_\psi(\hat{\mathbf{J}}_k, \hat{\mathbf{T}}_k), \tag{9}$$

where the $\hat{\mathbf{J}}_k$ we process by our SGDM can be presented as:

$$\hat{\mathbf{J}}_k = (\hat{\mathbf{T}}_k^\top \hat{\mathbf{T}}_k + \mu_k \mathbf{I})^{-1} \cdot (\hat{\mathbf{T}}_k^\top \mathbf{P} + \hat{\mathbf{T}}_k^\top \hat{\mathbf{T}}_k - \hat{\mathbf{T}}_k^\top + \mu_k \mathbf{J}_{k-1}), \tag{10}$$

as the $\lambda_k$ in transmission optimization, $\mu_k$ is also a learnable parameter to bring more generalization capabilities to the network.

**Details about CPMM.** T-CPMM and S-CPMM share the same structure for improved mapping quality. Each CPMM block uses a 4-channel convolution to embed $\mathbf{T}$ and $\mathbf{J}$ into a 30-dimensional feature map. The distinction between T-CPMM and S-CPMM lies in their outputs: T-CPMM produces a 1-channel result to aid TGDM in predicting a scene-compliant transmission map, whereas S-CPMM generates a 3-channel RGB image. This enables S-CPMM to learn additional scene feature information, such as atmospheric light and blur, assisting SGDM in generating higher-quality dehazed results with more details. For more efficient computation, each CPMM comprises only 3 layers with $[1, 1, 1]$ blocks, doubling the dimensions with increasing depth.

## 3.2 Coherence-based Pseudo Labeling by Colabator

We generate and select pseudo labels using our proposed plug-and-play coherence-based label generator, Colabator. Colabator consists of a teacher network with weights $\theta_{tea}$ shared with the student network $\theta_{stu}$ via exponential moving average (EMA). It employs a tailored mean-teacher strategy with a trust weighting process and an optimal label pool to generate high-quality pseudo labels, addressing the scarcity of real-world data. Figure 3 illustrates the pipeline of our Colabator.

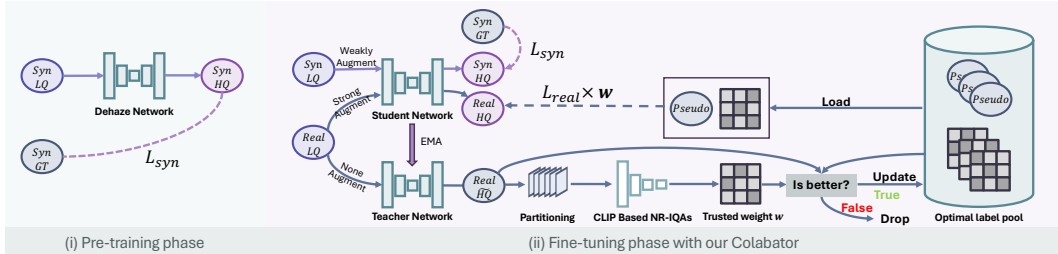

Figure 3: The plug-and-play Coherence-based Pseudo-label Generator paradigm.

**Iterative mean-teacher dehazing.** Given a real hazy image $\mathbf{P}_{LQ}^R \in \mathbb{R}^{H \times W \times 3}$, we initially apply augmentations to generate corresponding strongly degraded data using a strong augmentor $\mathcal{A}_s(\cdot)$, which randomly applies adjustments such as contrast, brightness, posterize, sharpness, JPEG compression, and Gaussian blur. Unlike the common mean-teacher strategy, we omit functions like solarize, equalize, shear, and translate to prevent unnecessary degradation that might mislead model learning. We use the non-augmented image as the input for the teacher network and the strongly augmented image for the student network, generating the following results:

$$\mathbf{P}_{\widetilde{HQ}}^R, \mathbf{T}_{\widetilde{HQ}}^R = f_{\theta_{tea}}(\mathbf{P}_{LQ}^R), \quad \mathbf{P}_{HQ}^R, \mathbf{T}_{HQ}^R = f_{\theta_{stu}}(\mathcal{A}_s(\mathbf{P}_{LQ}^R)), \tag{11}$$

where $\mathbf{P}_{\widetilde{HQ}}^R \in \mathbb{R}^{H \times W \times 3}$ is the result from the teacher network using the non-augmented input, and $\mathbf{P}_{HQ}^R \in \mathbb{R}^{H \times W \times 3}$ represents the result from the student network by strong augment input and $\mathbf{T}_{\widetilde{HQ}}^R$, $\mathbf{T}_{HQ}^R$ are the corresponding transmission map. The different degrees of data augmentation lead to varying dehazing results, typically resulting in $\mathbf{P}_{\widetilde{HQ}}^R$ having better quality than $\mathbf{P}_{HQ}^R$.

This approach ensures the model descends in the correct direction and helps mitigate the overfitting issues often associated with direct pseudo-label learning methods. By iterating, our teacher network generates increasingly high-quality pseudo-labels, providing more reliable supervision.

**Label trust weighting.** To better leverage the pseudo-dehazed images $\mathbf{P}_{\widetilde{HQ}}^R$ generated by the teacher network for model supervision, we designed a composite image quality assessment strategy for further processing these pseudo-dehazed images and get the trusted weight $w$ which means the reliability of each location of an image. Our composite strategy primarily consists of a haze density evaluator $\mathcal{D}(\cdot)$ based on pre-trained CLIP [38] model and fixed text feature, and a non-reference image quality evaluator $\mathcal{Q}(\cdot)$. We partition $\mathbf{P}_{\widetilde{HQ}}^R$ into an sequence $\mathbf{S}_{\widetilde{HQ}}^R \in \mathbb{R}^{N \times N \times 3 \times (H/N) \times (W/N)}$ and use $\mathcal{D}(\cdot)$ and $\mathcal{Q}(\cdot)$ to predict the density score and quality score. The final trusted weight $w$ we can get from:

$$w = \Psi(\text{norm}(\mathcal{D}(\mathbf{S}_{\widetilde{HQ}}^R)) \cdot \text{norm}(\mathcal{Q}(\mathbf{S}_{\widetilde{HQ}}^R))), \tag{12}$$

where $\Psi$ is compose sequence to map and resize as $\mathbf{P}_{\widetilde{HQ}}^R$, norm($\cdot$) means normalize scores from 0 to 1, that higher score means lower haze density and better image quality.

**Optimal label pool.** To ensure the use of optimal pseudo-labels and avoid domain adaptation collapse due to instability during training, we proposed an optimal label pool $\mathcal{P}$ to maintain the pseudo-labels in their optimal state. The overall procedure of our optimal label pool process is summarized in algorithm 1, compare pseudo-dehazed image $\mathbf{P}_{\widetilde{HQ}_i}^R$ with previous pseudo-label $\mathbf{P}_{Psei}^R$ and update pseudo-dehazed image as pseudo-label if it better than previous. To summarize the algorithm 1 and eq. (11), the overall process of Colabator can be formalize as:

$$\mathbf{P}_{HQ}^R, \mathbf{T}_{HQ}^R, \mathbf{P}_{Pse}^R, \mathbf{T}_{Pse}^R, w_{pse} = \mathcal{C}(\mathbf{P}_{LQ}^R, \theta_{tea}, \theta_{stu}, \mathcal{A}_s, \mathcal{D}(\cdot), \mathcal{Q}(\cdot), \mathcal{P}), \tag{13}$$

where $\mathcal{C}$ is our Colabator framework, $\mathbf{P}_{Pse}^R$ is the paired pseudo label of $\mathcal{A}_s(\mathbf{P}_{LQ}^R)$, $\mathbf{T}_{Pse}^R$ is the corresponding pesudo transmission map, $w_{pse}$ means the trusted weight of the pseudo label.

**Weights update.** The teacher network's weights $\theta_{tea}$ are updated by exponential moving average (EMA) of the student network's weights $\theta_{stu}$, which is denoted as follows:

$$\theta_{tea} = \eta\theta_{tea} + (1 - \eta)\theta_{stu}, \tag{14}$$

where $\eta$ is momentum and $\eta \in (0, 1)$. Using this update strategy, the teacher model can aggregate previously learned weights immediately after each training step, ensuring updating stability.

**Algorithm 1** Optimal label pool process

---

**Require:** Haze density evaluator $\mathcal{D}(\cdot)$ and image quality evaluator $\mathcal{Q}(\cdot)$;
Optimal label pool $\mathcal{P}$;
Sample a batch of real hazy images $\{\mathbf{P}_{LQ_i}^R\}_{i=1}^b$;
**for** each $\mathbf{P}_{LQ_i}^R$ **do**
    Get teacher network prediction: $\mathbf{P}_{\widetilde{HQ}_i}^R, \mathbf{T}_{\widetilde{HQ}_i}^R = f_{\theta_{tea}}(\mathbf{P}_{LQ_i}^R)$;
    Partition $\mathbf{P}_{\widetilde{HQ}_i}^R$ into $N \times N$ and get $\mathbf{S}_{\widetilde{HQ}_i}^R$;
    Compute score map of $\mathbf{S}_{\widetilde{HQ}_i}^R$: $d_i = \text{norm}(\mathcal{D}(\mathbf{S}_{\widetilde{HQ}_i}^R))$, and $q_i = \text{norm}(\mathcal{Q}(\mathbf{S}_{\widetilde{HQ}_i}^R))$;
    Load $\mathbf{P}_{Psei}^R, \mathbf{T}_{Psei}^R, w_{Psei}, d_{Psei}, q_{Psei} = \mathcal{P}(i)$
    **if** $d_i > d_{Psei}$ and $q_i > q_{Psei}$ **then**
        Compute trusted weight: $w_i = \Psi(d_i + q_i)$
        Update $\mathcal{P}(i) = (\mathbf{P}_{\widetilde{HQ}_i}^R, \mathbf{T}_{\widetilde{HQ}_i}^R, w_i, d_i, q_i)$
        Return $\mathbf{P}_{\widetilde{HQ}_i}^R, \mathbf{T}_{\widetilde{HQ}_i}^R, w_i$ as pesudo label.
    **else**
        Return $\mathbf{P}_{Psei}^R, \mathbf{T}_{Psei}^R, w_{pse_i}$ as pesudo label.
    **end if**
**end for**

---

## 3.3 Semi-supervised Real-world Image Dehazing

To achieve success in real-world dehazing, we designed several loss functions for our CORUN and Colabator to constrain their learning process. We introduce a reconstruction loss using the $L_1$ norm $\|\cdot\|_1$. To enhance visual perception, we employ contrastive and common perceptual regularization to ensure the consistency of the reconstruction results with the ground truth in terms of features at different levels. The perceptual loss is defined as follows:

$$L_{Rec}^{common}(\mathbf{P}_{HQ}, \mathbf{P}_{GT}) = \|\mathbf{P}_{GT}, \mathbf{P}_{HQ}\|_1 + \beta_c \sum_{i=1}^{n} \tau_i \|\varphi_i(\mathbf{P}_{GT}), \varphi_i(\mathbf{P}_{HQ})\|_1 \qquad (15)$$

$$L_{Rec}^{contra}(\mathbf{P}_{LQ}, \mathbf{P}_{HQ}, \mathbf{P}_{GT}) = \|\mathbf{P}_{GT}, \mathbf{P}_{HQ}\|_1 + \beta_c \sum_{i=1}^{n} \tau_i \frac{\|\varphi_i(\mathbf{P}_{GT}), \varphi_i(\mathbf{P}_{HQ})\|_1}{\|\varphi_i(\mathbf{P}_{LQ}), \varphi_i(\mathbf{P}_{HQ})\|_1}, \qquad (16)$$

where $\mathbf{P}_{HQ}$ is the dehazed result, $\varphi_i(\cdot)$ means the $i_{th}$ hidden layer of pre-trained VGG-19 [39], $\tau_i$ is the weight coefficient. Besides, to constrain the entire pipeline to obey physical laws while alleviating constraints on the intermediate layers, and prevent overfitting, we introduce a global coherence loss:

$$L_{Coh}(\mathbf{P}_{LQ}, \mathbf{P}_{HQ}, \mathbf{T}_{HQ}) = \|(\mathbf{P}_{HQ} \odot \mathbf{T}_{HQ} + (\mathbf{I} - \mathbf{T}_{HQ})) - \mathbf{P}_{LQ}\|_1, \qquad (17)$$

where $\odot$ is the Hadamard product, $\mathbf{I}$ means the all-ones matrix as the same size of $\mathbf{P}_{LQ}^S$. The global coherence loss ensures that CORUN can more efficiently integrate physical information into the deep network to facilitate the recovery of more physically consistent details. In addition, we introduce a density loss $L_{dens}$ based on $\mathcal{D}(\cdot)$ to score and constrain the model to dehaze in the semantic domain:

$$L_{Dens}(\mathbf{P}) = \mathcal{D}(\mathbf{P}). \qquad (18)$$

**Pre-training phase.** To ensure the capacity in dehazing and transmission map estimation, we pre-trained CORUN on synthetic paired datasets which contained clear image $\mathbf{P}_{GT}^S \in \mathbb{R}^{H \times W \times 3}$ and synthetic hazy image $\mathbf{P}_{LQ}^S \in \mathbb{R}^{H \times W \times 3}$. Setting $\mathbf{P}_{LQ}^S$ as input, we can get the result by

$$\mathbf{P}_{HQ}^S, \mathbf{T}_{HQ}^S = f_{\theta_{stu}}(\mathcal{A}_w(\mathbf{P}_{LQ}^S)), \qquad (19)$$

where $\mathcal{A}_w$ means weakly geometric data augment, $\mathbf{P}_{HQ}^S$ means the dehazed result of synthetic hazy image, and $\mathbf{T}_{HQ}^S$ is the corresponding transmission map. In the pre-training phase, our CORUN is

| Metrics | Hazy | PDN [11] | MBDN [14] | DH [15] | DAD [27] | PSD [19] | D4 [26] | RIDCP [7] | DGUN [10] | Ours |
|---|---|---|---|---|---|---|---|---|---|---|
| FADE↓ | 2.484 | 0.876 | 1.363 | 1.895 | 1.130 | 0.920 | 1.358 | 0.944 | 1.111 | 0.824 |
| BRISQUE↓ | 36.642 | 30.811 | 27.672 | 33.862 | 32.241 | 27.713 | 33.210 | 17.293 | 27.968 | 11.956 |
| NIMA↑ | 4.483 | 4.464 | 4.529 | 4.522 | 4.312 | 4.598 | 4.484 | 4.965 | 4.653 | 5.342 |

Table 1: Quantitative results on RTTS dataset. Red and blue indicate the best and the second best.

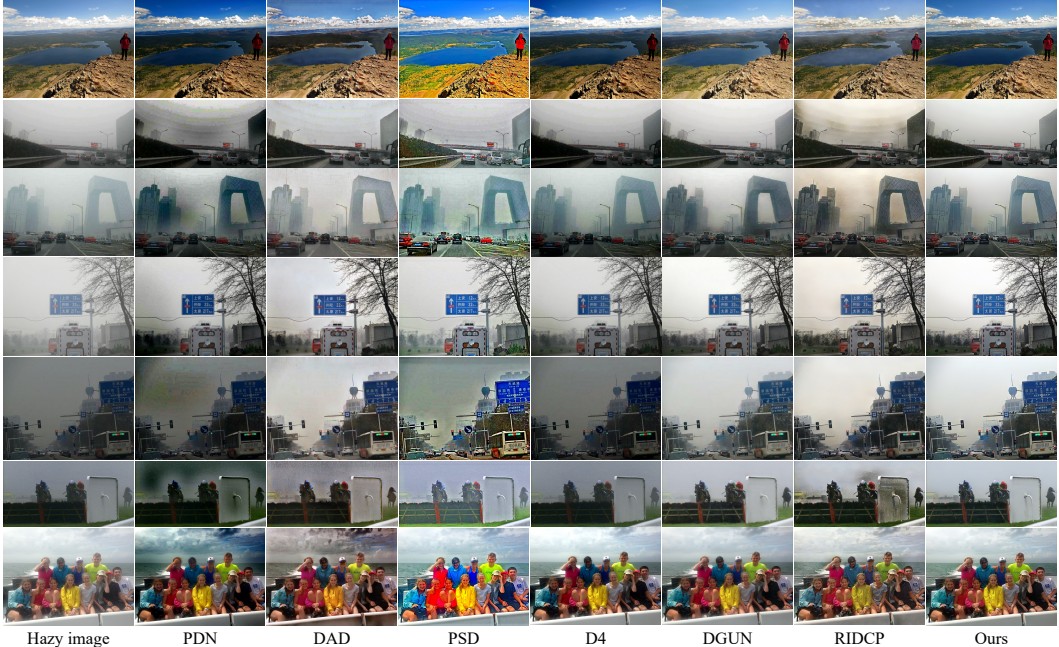

| Hazy image | PDN | DAD | PSD | D4 | DGUN | RIDCP | Ours |

Figure 4: Visual comparison on RTTS[40]. Please zoom in for a better view.

optimized end-to-end using two supervised loss functions. The overall loss of the pre-training phase:

$$L_{pre} = \rho_r L_{Rec}^{contra}(\mathcal{A}_w(\mathbf{P}_{LQ}^S), \mathbf{P}_{HQ}^S, \mathbf{P}_{GT}^S)$$
$$+ \rho_c L_{Coh}(\mathcal{A}_w(\mathbf{P}_{LQ}^S), \mathbf{P}_{HQ}^S, \mathbf{T}_{HQ}^S) + L_{Dens}(\mathbf{P}_{HQ}^S), \tag{20}$$

where $\rho_r$ is the trade-off weight of $L_{Rec}^{contra}$, $\rho_c$ is the trade-off weight of $L_{Coh}$.

**Fine-tuning phase.** In fine-tuning phase, we adapt our CORUN pre-trained on synthetic data to the real-world domain by our Colabator framework. For more steady learning, in this phase, we train with both synthetic and real-world data. As eq. (13), we generate $\mathbf{P}_{HQ}^R, \mathbf{T}_{HQ}^R, \mathbf{P}_{Pse}^R, \mathbf{T}_{Pse}^R, w_{pse}$ from $\mathbf{P}_{LQ}^R$, and we get $\mathbf{P}_{HQ}^S, \mathbf{T}_{HQ}^S$ use the eq. (19). The overall loss of the fine-tuning phase:

$$L_{fine} = w\rho_r L_{Rec}^{contra}(\mathcal{A}_s(\mathbf{P}_{LQ}^R), \mathbf{P}_{HQ}^R, \mathbf{P}_{Pse}^R) + \rho_r L_{Rec}^{common}(\mathbf{P}_{HQ}^S, \mathbf{P}_{GT}^S)$$
$$+ w\rho_c L_{Coh}(\mathcal{A}_s(\mathbf{P}_{LQ}^R), \mathbf{P}_{HQ}^R, \mathbf{T}_{HQ}^R) + L_{Dens}(\mathbf{P}_{HQ}^S) + wL_{Dens}(\mathbf{P}_{HQ}^R). \tag{21}$$

## 4 Experiments

### 4.1 Experimental Setup

**Data Preparation.** We use RIDCP500 [7] dataset, comprising 500 clear images with depth maps estimated by [41], and follow the same way of RIDCP [7] for generating paired data. During the fine-tuning phase, we incorporate the URHI subset of RESIDE dataset [40], which only consists of 4,807 real hazy images, for generating pseudo-labels and fine-tuning the network. We evaluate our framework qualitatively and quantitatively on the RTTS subset, which comprises over 4,000 real hazy images featuring diverse scenes, resolutions, and degradation. Fattal's dataset [42], comprising 31 classic real hazy cases, serves as a supplementary source for cross-dataset visual comparison.

**Implementation Details.** Our framework is implemented using PyTorch [43] and trained on four NVIDIA RTX 4090 GPUs. During the pre-training phase, we train the network for 30K iterations,

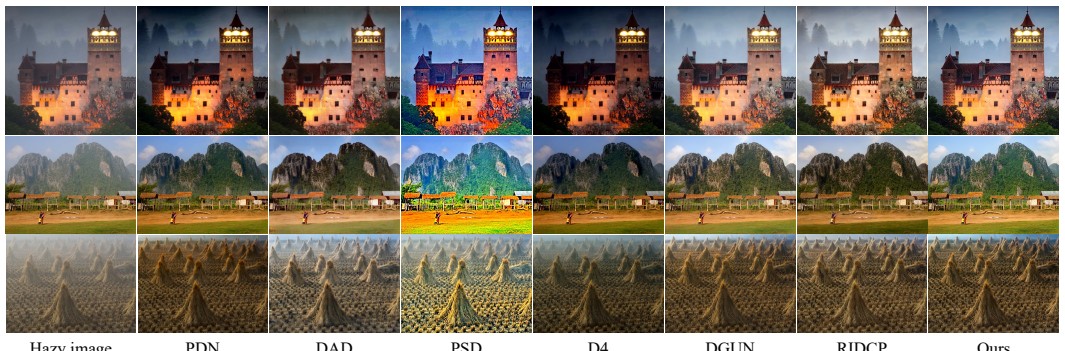

| Hazy image | PDN | DAD | PSD | D4 | DGUN | RIDCP | Ours |

Figure 5: Visual comparison on Fattal's data[42].

| Datasets | Metrics | w/o Colabator DGUN | w/ Colabator DGUN | w/o Colabator CORUN | w/ Colabator CORUN (Ours) |
|---|---|---|---|---|---|
| RTTS | FADE↓ | 1.111 | 0.857 | 1.091 | 0.824 |
| | BRISQUE↓ | 25.085 | 20.731 | 16.541 | 11.956 |
| | NIMA↑ | 4.813 | 5.190 | 4.856 | 5.342 |

Table 2: Generalization and Effect of our Colabator.

| Datasets | Metrics | w/o Mean-teacher | w/o Trusted weight | w/o Optimal label pool |
|---|---|---|---|---|
| RTTS | FADE↓ | 0.912 | 0.827 | 0.846 |
| | BRISQUE↓ | 15.728 | 16.606 | 15.707 |
| | NIMA↑ | 4.921 | 4.867 | 5.285 |

Table 3: Module's Effect of our Colabator.

| Datasets | Metrics | Stages 1 | 2 | 4 (Ours) | 6 |
|---|---|---|---|---|---|
| RTTS | FADE↓ | 0.785 | 0.808 | 0.824 | 0.839 |
| | BRISQUE↓ | 15.520 | 15.151 | 11.956 | 16.227 |
| | NIMA↑ | 5.228 | 5.281 | 5.342 | 5.187 |

Table 4: Effect of stage number.

optimizing it with AdamW [44] using momentum parameters ($\beta_1 = 0.9, \beta_2 = 0.999$) and an initial learning rate of $2 \times 10^{-4}$, gradually reduced to $1 \times 10^{-6}$ with cosine annealing. In Colabator, the initial learning rate is set to $5 \times 10^{-5}$ with only 5K iterations. Following [7], we employ random crop and flip for synthetic data augmentation. We use DA-CLIP [45] as our haze density evaluator and MUSIQ [46] as the image quality evaluator. Our CORUN consists of 4 stages and the trade-off parameters in the loss are set to $\beta_c, \rho_r, \rho_c$ are set to $0.2, 5, 10^{-2}$, respectively.

**Metrics.** We utilize the Fog Aware Density Evaluator (FADE) [47] to assess the haze density in various methods. However, FADE focuses on haze density exclusively, overlooking other crucial image characteristics such as color, brightness, and detail. To address this limitation, we also employ Blind/Referenceless Image Spatial Quality Evaluator (BRISQUE) [48], and Neural Image Assessment(NIMA) [49] for a more comprehensive evaluation of image quality and aesthetic. Higher NIMA scores, along with lower FADE and BRISQUE scores, indicate better performance. We use PyIQA [50] for BRISQUE and NIMA calculations, and the official MATLAB code for FADE calculations. All of these metrics are non-reference because there is no ground-truth in RTTS [40].

## 4.2 Comparative Evaluation

We compare our method with 8 state-of-the-art methods: PDN [11], MBDN [14], DH [15], DAD [27], PSD [19], D4 [26], RIDCP [7], DGUN [10]. The quantitative results, presented in table 1, show that our method achieved the highest performance, outperforming the second-best method (RIDCP) by 17.0%. Specifically, our method improved FADE, BRISQUE, and NIMA scores by 12.7%, 30.8%, and 7.6%, respectively. This demonstrates that our method surpasses current state-of-the-art techniques in both dehazing capability and the quality, and aesthetics of the generated images.

The visual comparisons of our proposed method and state-of-the-art algorithms are shown in figs. 4 and 5. We can observe that these methods have demonstrated some effectiveness in real-world dehazing tasks, but when images containing white objects, sky, or extreme haze, the results from PDN, DAD, PSD, and RIDCP exhibited varying degrees of dark patches and contrast inconsistencies. Conversely, D4 caused an overall reduction in brightness, leading to detail loss in darker areas. Under these conditions, DGUN produced relatively aesthetically pleasing results but lost significant local detail, impairing overall visual quality. Notably, PSD achieved higher brightness but suffered from

| Dataset | PDN [11] | MBDN [14] | DH [15] | DAD [27] | PSD [19] | D4 [26] | RIDCP [7] | DGUN [10] | Ours |
|---|---|---|---|---|---|---|---|---|---|
| RTTS[40] | 4.52 | 3.47 | 3.23 | 4.35 | 3.90 | 4.66 | 7.14 | 6.04 | 7.76 |
| Fattal's[42] | 4.85 | 3.33 | 3.19 | 4.80 | 4.28 | 4.38 | 7.28 | 6.33 | 8.04 |

Table 5: User study scores on RTTS[40] and Fattal's[42].

| Class(AP) | Hazy | PDN [11] | MBDN [14] | DH [15] | DAD [27] | PSD [19] | D4 [26] | RIDCP [7] | DGUN [10] | Ours |
|---|---|---|---|---|---|---|---|---|---|---|
| Bicycle | 0.51 | 0.55 | 0.54 | 0.47 | 0.52 | 0.52 | 0.54 | 0.57 | 0.55 | 0.59 |
| Bus | 0.25 | 0.29 | 0.27 | 0.23 | 0.29 | 0.25 | 0.28 | 0.32 | 0.31 | 0.31 |
| Car | 0.61 | 0.65 | 0.63 | 0.51 | 0.65 | 0.63 | 0.64 | 0.67 | 0.66 | 0.68 |
| Motor | 0.38 | 0.45 | 0.43 | 0.37 | 0.38 | 0.42 | 0.42 | 0.47 | 0.46 | 0.49 |
| Person | 0.73 | 0.76 | 0.75 | 0.69 | 0.74 | 0.74 | 0.75 | 0.76 | 0.76 | 0.77 |
| Mean | 0.50 | 0.54 | 0.52 | 0.45 | 0.52 | 0.51 | 0.53 | 0.56 | 0.55 | 0.57 |

Table 6: Object detection results on RTTS[40].

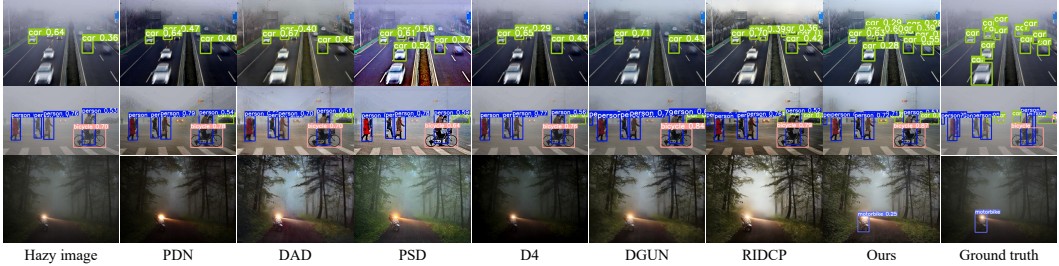

Hazy image    PDN    DAD    PSD    D4    DGUN    RIDCP    Ours    Ground truth

Figure 6: Visual comparison of object detection on RTTS [40].

severe oversaturation. CORUN+ consistently outperforms others by producing clearer images with natural colors and better contrast, effectively removing haze while preserving image details.

### 4.3 Ablation Study

**Generalization and Effect of Colabator.** We evaluates the performance and the impact of our proposed Colabator framework across different metrics. As shown in table 2, removing the fine-tuning phase of Colabator led to significant performance drops, highlighting its critical role in the dehazing process. To evaluate the generalizability of Colabator, we conducted additional experiments by replacing our CORUN with the DGUN [10], while maintaining consistent training settings. Results in table 2 and fig. 1 indicate that Colabator substantially enhances DGUN's performance, demonstrating its effectiveness as a plug-and-play paradigm with strong generalization capabilities.

**Effect of Colabator.** We validate the effect of our Colabator. In table 3, we systematically removed critical components, such as mean-teacher, trusted weighting, and the optimal label pool, from the model architecture. The outcomes indicate the performance deteriorates when these components are removed, highlighting their essential role in the system.

**Ablations on stage number.** The number of stages in a deep unfolding network significantly impacts its efficiency and performance. To investigate this, we experimented with different stage numbers for CORUN+, specifically choosing $k$ values from the set $\{1, 2, 4, 6\}$. The results detailed in table 4, indicate that CORUN+ achieves high-quality dehazing with 4 stages. Notably, increasing the number of stages does not necessarily improve outcomes. Excessive stages can increase the network's complexity, hinder convergence, and potentially introduce errors in the results.

### 4.4 User Study and Downstream Task

**User Study.** We conducted a user study to evaluate the human subjective visual perception of our proposed method against other methods. We invited five experts with an image processing background and 16 naive observers as testers. These testers were instructed to focus on three primary aspects: (i) Haze density compared to the original hazy image, (ii) Clarity of details in the dehazed image, and (iii) Color and aesthetic quality of the dehazed image. The results for each method, along with the corresponding hazy images, were presented to the testers anonymously. They scored each method on a scale from 1 (worst) to 10 (best). The hazy images were selected randomly, with a total of 225 images from RTTS[51] and 54 images from Fattal's[42] dataset. The user study scores are reported in table 5, showing that our method achieved the highest average score.

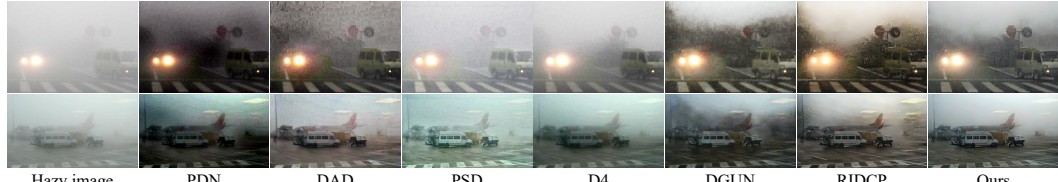

| Hazy image | PDN | DAD | PSD | D4 | DGUN | RIDCP | Ours |

Figure 7: Failure cases. Our results show low quality texture details.

**Downstream Task Evaluation.** The performance of high-level vision tasks, *e.g.* object detection and semantic segmentation, is greatly affected by image quality, with severely degraded images often leading to erroneous results [52, 53]. To address this performance degradation, some methods have incorporated image restoration as a preprocessing step for high-level vision tasks. To validate the effectiveness of our approach for high-level vision, we utilized pretrained YOLOv3 [54], and tested it on the RTTS [40] dataset, and evaluated the results using the mean Average Precision (mAP) metric. As shown in table 6 and fig. 6, our method demonstrates a substantial advantage over existing methods, verifying our efficacy in facilitating high-level vision understanding.

# 5  Limitations and Future Work

In fig. 7, our CORUN+ model struggles to maintain result quality and preserve texture details when dealing with severely degraded inputs, such as strong compression and extreme high-density haze. This challenge persists across existing methods and remains unresolved. We attribute this difficulty to the model's struggle in reconstructing scenes from dense haze, where information is often severely lacking or entirely lost, affecting the reconstruction of both haze-free and low haze density areas. Moreover, the model solely focuses on dehazing and lacks the capability to address other image degradations, such as image deblurring[55] and low-light image enhancement [56, 57], limiting its ability to achieve high-quality reconstruction results from complex degraded images. To address this limitation in future research, we propose not only focusing on environmental degradation but also considering additional information about image degradation when solving real-world dehazing problems. In addition to this, we can integrate robust generative methods to improve the network's ability to restore dense haze regions [58–62], synthesize haze that matches real-world distributions [63–66], and introduce more modalities as supplements to RGB images, enhancing the model's ability to effectively recover details [67].

# 6  Conclusions

In this paper, we introduce CORUN to cooperatively model atmospheric scattering and image scenes and thus incorporate physical information into deep networks. Furthermore, we propose Colabator, an iterative mean-teacher framework, to generate high-quality pseudo-labels by storing the best-ever results with global and local coherence in a dynamic label pool. Experiments demonstrate that our method achieves state-of-the-art performance in real-world image dehazing tasks, with Colabator also improving the generalization of other dehazing methods. The code will be released.

## Acknowledgments and Disclosure of Funding

This work was supported by the STI 2030-Major Projects under Grant 2021ZD0201404. The authors thank the NeurIPS committee for granting us the NeurIPS 2024 Scholar Award, which has helped us participate in the conference.

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

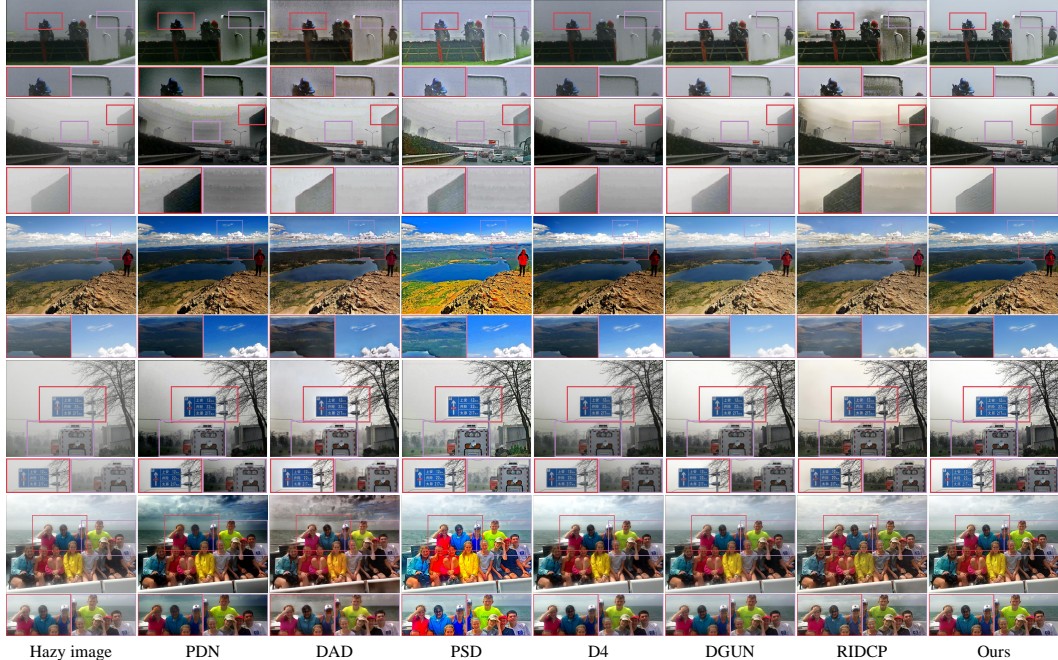

Figure 8: Detailed Comparison of fig. 4. **Red** region reflects all past methods have had haze residues, but our method have the least in the same case. **Purple** region shows our method restores richer detail and truer colors. Please zoom in for a better view.

## A  Appendix

### A.1  Declaration of eq. (7) and eq. (10)

Having gotten eq. (5) and eq. (6), the solution of $\hat{\mathbf{T}}$ can be formulated according to the proximal gradient algorithm:

$$\mathcal{T}(\hat{\mathbf{J}}_{k-1}, \mathbf{T}_{k-1}) = \frac{1}{2}\|\mathbf{P} - \hat{\mathbf{J}}_{k-1} \cdot \hat{\mathbf{T}} + \hat{\mathbf{T}} - \mathbf{I}\|_2^2 + \frac{\lambda_k}{2}\|\hat{\mathbf{T}} - \mathbf{T}_{k-1}\|_2^2. \tag{22}$$

Then we obtain the partial derivative:

$$\partial_{\hat{\mathbf{T}}}\mathcal{T}(\hat{\mathbf{J}}_{k-1}, \mathbf{T}_{k-1}) = (\mathbf{I} - \hat{\mathbf{J}}_{k-1})^T(\mathbf{P} - \hat{\mathbf{J}}_{k-1} \cdot \hat{\mathbf{T}} + \hat{\mathbf{T}} - \mathbf{I}) + \lambda_k(\hat{\mathbf{T}} - \mathbf{T}_{k-1}). \tag{23}$$

Let the partial derivative be equal to zero, we archieve the closed-form solution for $\hat{\mathbf{T}}$ in eq. (7).

Similarly, the solution of $\hat{\mathbf{J}}$ can be formulated as

$$\mathcal{J}(\hat{\mathbf{T}}_k, \mathbf{J}_{k-1}) = \frac{1}{2}\|\mathbf{P} - \hat{\mathbf{J}} \cdot \hat{\mathbf{T}}_k + \hat{\mathbf{T}}_k - \mathbf{I}\|_2^2 + \frac{\mu_k}{2}\|\hat{\mathbf{J}} - \mathbf{J}_{k-1}\|_2^2. \tag{24}$$

The corresponding partial derivative is

$$\partial_{\hat{\mathbf{J}}}\mathcal{J}(\hat{\mathbf{T}}_k, \mathbf{J}_{k-1}) = -\hat{\mathbf{T}}_k^T(\mathbf{P} - \hat{\mathbf{J}} \cdot \hat{\mathbf{T}}_k + \hat{\mathbf{T}}_k - \mathbf{I}) + \mu_k(\hat{\mathbf{J}} - \mathbf{J}_{k-1}). \tag{25}$$

The closed-form solution for $\hat{\mathbf{J}}$ is presented in eq. (10) when let the partial derivative be equal to zero.

### A.2  Declaration of CLIP module.

The haze density evaluator $\mathcal{D}(\cdot)$ can be formulated as:

$$\mathcal{D}(\cdot) = \frac{Enc_{image}(\cdot)}{\|Enc_{image}(\cdot)\|} \cdot \left(\frac{Enc_{text}(\text{Text})}{\|Enc_{text}(\text{Text})\|}\right)^\top \tag{26}$$

The text we used is "hazy" from the DA-CLIP provided in the text list.

Table 7: Ablation of CPMM module of our CORUN.

| Modules | NIMA ↑ | BRISQUE ↓ | FADE ↓ |
|---|---|---|---|
| Hazy(Input) | 4.483 | 36.642 | 2.484 |
| w/o CPMM | 4.836 | 38.197 | 1.362 |
| w/ CPMM (CORUN+) | 5.342 | 11.956 | 0.824 |

Table 8: Ablation of our trusted weights present as a map or value.

| Methods | NIMA ↑ | BRISQUE ↓ | FADE ↓ |
|---|---|---|---|
| Only Full | 5.229 | 13.099 | 0.803 |
| Partition+Full(CORUN+) | 5.342 | 11.956 | 0.824 |

Table 9: Effects of more Colabator components.

| Modules | NIMA ↑ | BRISQUE ↓ | FADE ↑ |
|---|---|---|---|
| w/o Colabator | 4.856 | 16.541 | 1.091 |
| w/o Strong aug. | 5.084 | 12.671 | 0.813 |
| w/o DA-CLIP[45] | 5.358 | 11.200 | 0.856 |
| CORUN+ | 5.342 | 11.956 | 0.824 |

Table 10: Ablation of mainstream datasets setting.

| RIDCP[7] Pipeline | | - | | Metrics | | |
|---|---|---|---|---|---|---|
| Data.+Gen. | Aug. | OTS | NIMA ↑ | BRISQUE ↓ | FADE↓ |
| ✓ | | | | 4.845 | 20.779 | 0.765 |
| | | ✓ | | 4.991 | 16.478 | 0.840 |
| ✓ | ✓ | | | 5.342 | 11.956 | 0.824 |

## A.3 Ablation Study

**Effect of the CPMM Module in CORUN.** We evaluate the impact of the Cooperative Proximal Mapping Modules (CPMM) in our CORUN architecture. As shown in table 7, incorporating CPMM leads to a significant improvement in performance across all metrics. The model with CPMM (CORUN+) outperforms the variant without CPMM, highlighting the importance of CPMM in enhancing dehazing efficiency and image quality.

**Impact of Trusted Weight Representations.** We investigate the effect of using trusted weights as either a map or a value. As seen in table 8, the combination of partitioned and full trusted weights (CORUN+) achieves better results compared to using only full trusted weights. This emphasizes the value of our trusted weight representation in improving the accuracy of dehazing.

**Effects of Additional Colabator Components.** To analyze the contribution of individual components within the Colabator framework, we performed ablation studies, with the results presented in table 9. Removing essential elements, such as strong augmentation or DA-CLIP, results in performance deterioration, confirming the importance of each Colabator component in ensuring optimal dehazing outcomes.

**Ablation on Dataset Configurations.** We evaluate our methods with different dataset settings. As shown in table 10, the results verify our method still achieves a leading place under the three settings compared with existing methods.

**Effectiveness of the Simplified ASM Formula.** We assess the impact of simplifying the Atmospheric Scattering Model (ASM) formula. The results in table 11 indicate that using the simplified ASM formula will lead to a slight decrease in the dehazing ability, but it can evidently improve the image quality of the results.

**Influence of Loss Functions.** We compare the effect of using different loss functions (eq. (15) and eq. (16)). Table 12 shows that combining both loss functions yields better performance than using either one alone, demonstrating the advantage of this combined loss strategy in refining the dehazing process.

Table 11: Ablation of our simplified ASM formula.

| ASM formula | NIMA ↑ | BRISQUE ↓ | FADE↓ |
|---|---|---|---|
| w/o simplify | 5.203 | 14.469 | 0.817 |
| w/ simplify(CORUN+) | 5.342 | 11.956 | 0.824 |

Table 12: Ablations of eq.15 and eq.16 loss functions. Our strategy achieves a better result.

| Loss | NIMA ↑ | BRISQUE ↓ | FADE ↓ |
|---|---|---|---|
| Eq.15 Only | 5.249 | 13.997 | 1.035 |
| Eq.16 Only | 5.220 | 12.484 | 0.795 |
| Both Loss (Ours) | 5.342 | 11.956 | 0.824 |

Table 13: Effects of integrating our Colabator with more cutting-edge dehazing methods. The gains brought by Colabator are significant.

| Methods | NIMA ↑ | BRISQUE ↓ | FADE↓ |
|---|---|---|---|
| C2PNet[22] | 4.715 | 34.314 | 2.064 |
| C2PNet+Colabator | 4.823 | 23.662 | 1.329 |
| FFA-Net[17] | 4.822 | 33.235 | 2.080 |
| FFA-Net+Colabator | 4.839 | 29.219 | 0.958 |
| GDN[16] | 5.074 | 33.051 | 2.611 |
| GDN+Colabator | 5.258 | 23.691 | 0.947 |

**Integration with Other Dehazing Methods.** To test the generalizability of Colabator, we integrated it with various state-of-the-art dehazing models, such as C2PNet [22], FFA-Net [17], and GDN [16]. As shown in table 13, incorporating Colabator leads to performance gains across all metrics, demonstrating its effectiveness as a plug-and-play module for improving dehazing in various architectures.

## A.4 Broader Impacts

Real-world image dehazing is a crucial task in image restoration, aimed at removing haze degradation from images captured in real-world scenarios. In computer vision, dehazing can benefit downstream tasks such as object detection [68–72], image segmentation [73–81], tracking [82, 83], depth estimation [84, 85], and more vision related tasks [86–94], with applications ranging from autonomous driving to security monitoring. Our paper introduces a cooperative unfolding network and a plug-and-play pseudo-labeling framework, achieving state-of-the-art performance in real-world dehazing tasks. Notably, image dehazing techniques have yet to exhibit negative social impacts. Our proposed CORUN and Colabator methods also do not present any foreseeable negative societal consequences.

