# OpenReview forum: "Real-world Image Dehazing with Coherence-based Pseudo Labeling and Cooperative Unfolding Network"
_NeurIPS.cc/2024/Conference — NeurIPS 2024 spotlight_

### Official Review · Reviewer_wF18 · 2024-07-01

**Soundness:** 3
**Presentation:** 3
**Contribution:** 3
**Rating:** 8
**Confidence:** 5

**Summary:**

This paper posits that real-world image dehazing is particularly challenging due to the intricacies of accurately modeling haze distributions and the limited availability of paired real-world data. Traditional and deep learning-based methods struggle to address the complexities of real haze, often resulting in color distortion and suboptimal outcomes. To tackle these issues, this paper introduces the CORUN model, which integrates atmospheric scattering and image scenes to incorporate physical information into deep networks. Furthermore, it proposes the Coherence-based Label Generator to produce high-quality pseudo labels for network training, thereby improving the network's generalization in haze removal.

**Strengths:**

1. The proposed method, CORUN (COopeRative Unfolding Network), integrates physical information into deep networks to overcome the limitations of existing deep learning-based dehazing methods. It cooperatively models atmospheric scattering and image scenes by incorporating Transmission and Scene Gradient Descent Modules at each stage, effectively restoring haze-contaminated details.
2. CORUN is constructed on the basis of the atmospheric scattering model using a proximal gradient descent method within a deep unfolding network, which provides strong interpretability to the method.
3. The proposed Colabator framework significantly improves the performance of models pretrained on synthetic datasets in real-world scenarios within a limited number of iterations by fine-tuning with real degraded images. This plug-and-play framework incurs no additional computational cost during deployment.
4. Experiments demonstrate the excellent performance of the proposed CORUN model and Colabator framework in real-world dehazing tasks. Both quantitative and visual results validate the network's capability to model real-world haze and restore real-world scenes effectively.

**Weaknesses:**

1. The paper contains some inconsistencies in the use of symbols. While images are generally denoted by P, in the loss functions (15, 16, 17), the authors use I to represent images. Although the text explains the use of these symbols, it can still impact readability. Additionally, in line 172, the pseudo-label symbol is misspelled and should be corrected to P^{R}\_{Pse\_{i}}.
2. In line 175, the symbols contain unnecessary tildes, which should be removed. The symbols in Figure 2 should also be consistent with those used in the formulas.

3. Equations (5) and (7) do not address the consistency of the number of channels between the Transmission Map and the image. Although this is illustrated in Figure 2, it still needs to be clearly stated in the equations to facilitate readability.

4. This paper adequately demonstrates its method's performance in real-world dehazing tasks through thorough experiments in both visualization and quantitative results. However, the provided ablation studies are insufficient. The authors should conduct more comprehensive ablation studies and experiments on different datasets to further validate the effectiveness of their method and its components.

**Questions:**

1. This paper introduce Colabator, a plug-and-play coherence label generation method designed to fine-tune models pre-trained on synthetic datasets using real degraded images, thereby achieving better real-world image processing results. Can the authors provide additional results of this framework's fine-tuning on various tasks to support this claim?
2. I noticed that the authors assign weights to the local quality of pseudo-labels using a weighted combination of CLIP scores and NR-IQA scores, instead of the more common approach of using their dot product. Can the authors provide experimental results to demonstrate the effectiveness of this choice?
3. Regarding the choice of NR-IQA, why did this paper select MUSIQ over other methods? What are the reasons behind this decision?
4. There is a small portion of similar or identical image content between the RTTS and URHI subsets of the RESIDE dataset. Did the authors notice this issue during their experiments? Have they addressed this issue, or did they directly use the full URHI subset for fine-tuning? If the authors used the entire URHI subset for fine-tuning, have they considered the model's performance on the RTTS subset after removing these similar images?

**Limitations:**

Regarding limitations, the proposed method partially addresses previous issues such as the lack of paired real-world data and the difficulty in modeling complex real-world haze distributions. Section A.1 "Limitations and Future Work" in the appendix discusses the current work's limitations and potential future research directions to address these unresolved issues. As for broader impacts, Section A.2 in the appendix provides a thorough discussion. As stated in the paper, this work has no foreseeable negative impacts on the field or society and offers substantial positive contributions.

---

> ### Author Rebuttal · Authors · 2024-08-06
>
> Thanks for the valuable comments. If not specifically stated, all experiments are conducted on the real-world image dehazing task with *RTTS* for space limitation.
>
> **W1 & W2: Typo and Misuse of symbols.**
>
> We apologize for the errors in our writing and appreciate you pointing out these mistakes in our article. We will correct the issues you mentioned in the final version to improve the readability of the paper.
>
> **W3: Lack of clarity in the formulas**
>
> We have revised Eq. (5) and Eq. (7) to clearly express the handling of information across different channels. The updated Eq. (5) is as follows:
>
> $$
> \mathbf{T}\_k=\underset{\mathbf{T}}{\arg \min }\frac{1}{2}\sum\_{c\in\{R,G,B\}} \| \mathbf{P}^{c} - \hat{\mathbf{J}}\_{k-1}^{c}\cdot \mathbf{T} + \mathbf{T} - \mathbf{I} \|^{2}\_{2}+\phi(\mathbf{T}).
> $$
>
> The updated Eq. (7) is as follows:
>
> $$
> \hat{\mathbf{T}}\_k=\sum\_{c\in\{R,G,B\}}(\mathbf{I}-\hat{\mathbf{J}}\_{k-1}^{c}+\frac{\lambda\_{k}}{(\mathbf{I}-\hat{\mathbf{J}}\_{k-1}^{c})^{\top}})^{-1} \cdot (\mathbf{I}-\mathbf{P}^{c}\frac{\lambda\_{k} \mathbf{T}\_{k-1}}{(\mathbf{I} -\hat{\mathbf{J}}\_{k-1}^{c})^{\top}}).
> $$
>
> We will update this content in the final version.
>
> **W4: More Ablation studies.**
>
> We tested our CORUN method on the O-HAZE and I-HAZE datasets, using PSNR and SSIM for evaluation. The results, presented in **Table A4**, illustrate the robustness of CORUN across dehazing datasets.
>
> **Q1: Plug-in-play experiment**
>
> Thank you for your suggestion. According to our experiments, Colabator exhibits plug-and-play characteristics and brings enhancements across different networks and real-world tasks. In **Figure. 1** and **Table. 2** of the paper, we validated the effectiveness of Colabator on deep unfolding networks and real-world image dehazing tasks. Here, we integrate our Colabator with more cutting-edge image restoration methods and fine-tune them 5000 iterations in *underwater image enhancement* and *Real robotic laparoscopic hysterectomy desmoke tasks after pretraining*.
>
> In the UIE task, all methods pretrained on the *UIEB* dataset, finetuned and tested on the *EUVP* trainset’s low-quality part and testset, we find the performance average gain by **11.3%** in SwinIR [68], **20.6%** by Restormer[69], **15.4%** by GRL[70], and **13.0%** by AST[71].
>
> ||NIMA↑|BRISQUE↓|FADE↓|
> |---|---|---|---|
> |SwinIR[68]|3.392|31.775|0.536|
> |SwinIR+Colabator|3.650|27.247|0.472|
> |Restormer[69]|3.981|25.688|0.482|
> |Restormer+Colabator|4.185|16.489|0.381|
> |GRL[70]|3.816|25.795|0.506|
> |GRL+Colabator|3.925|18.854|0.423|
> |AST[71]|4.073|22.772|0.473|
> |AST+Colabator|4.186|16.353|0.435|
>
> In the Desmoke task, all methods pretrained on *Desmoke-LAP* with synthetic paired data, finetuned and tested on *Desmoke-LAP*’s real unpaired data, we find the performance average gain by **7.9%** in SwinIR [68], **4.9%** by Restormer[69], **7.1%** by GRL[70], and **6.4%** by AST[71].
>
> ||NIMA↑|BRISQUE↓|FADE↓|
> |---|---|---|---|
> |SwinIR[68]|3.056|40.668|0.655|
> |SwinIR+Colabator|3.317|35.519|0.639|
> |Restormer[69]|3.554|33.356|0.628|
> |Restormer+Colabator|3.738|30.933|0.614|
> |GRL[70]|3.503|31.682|0.601|
> |GRL+Colabator|3.716|27.663|0.585|
> |AST[71]|3.808|29.735|0.563|
> |AST+Colabator|3.952|26.483|0.538|
>
> The gains brought by our Colabator are significant, demonstrating that Colabator possesses plug-and-play capabilities across different networks and tasks.
>
> **Q2: Ablation study on trusted weights calculation**
>
> We apologize for not explicitly explaining the reason for using weighted summation in the text. We opted for summation rather than multiplication to calculate the Trusted weight, primarily because both the CLIP score and NR-IQAs score need to contribute their respective effects to the weight distribution. This approach also allows for the adjustment of different score ratios for different tasks to achieve more reasonable and better results. If weighted multiplication were used, it would be impossible to control the importance between different components. As shown in the table, we experimentally demonstrated that using the score summation method to calculate the Trusted weight, compared to using score multiplication, results in better final outcomes.
>
> ||NIMA↑|BRISQUE↓|FADE↓|
> |---|---|---|---|
> |Product|5.487|14.949|0.807|
> |Summation (Ours)|5.315|11.956|0.751|
>
> **Q3: Ablation study on NR-IQA**
>
> Thank you for your question. In fact, we have previously tested many NRIQA methods and found that MUSIQ has the best alignment with the human visual system. MUSIQ evaluates images at multiple scales, comprehensively considering both the technical and aesthetic quality of images, making it broadly applicable to various image quality assessment tasks. We conducted ablation experiments using different NR-IQA methods, and the results show that using MUSIQ in the scenarios presented in this paper can achieve better overall performance.
>
> ||NIMA↑|BRISQUE↓|FADE↓|
> |---|---|---|---|
> |NIMA|5.337|12.242|0.825|
> |BRISQUE|5.245|11.652|0.813|
> |NIQE|5.122|12.900|0.835|
> |MUSIQ(Ours CORUN+)|5.315|11.956|0.751|
>
> **Q4: Similar portion in RTTS and URHI**
>
> Thank you for your question. We used the URHI dataset following the common setting as referenced in works like PSD, so the issue does not cause unfairness in the comparative experiments. However, this concern is worth considering. Therefore, we attempted to remove images from the URHI dataset that overlap more than 60% with the RTTS dataset, ultimately retaining 3728 images for the second phase of fine-tuning CORUN. The final results are shown below. From the results, it can be seen that our method still achieves excellent performance even after using the cleaned dataset.
>
> We found that the NIMA score actually increased, which we believe is due to the removal of some low-quality data during the data cleaning process.
>
> ||NIMA↑|BRISQUE↓|FADE↓|
> |---|---|---|---|
> |URHI-|5.348|12.784|0.832|
> |URHI|5.315|11.956|0.751|
>
> [68]SwinIR, ICCVW, 2021
>
> [69]Restormer, CVPR, 2022
>
> [70]GRL, CVPR, 2023
>
> [71]AST, CVPR, 2024

---

> > ### Comment · Reviewer_wF18 · 2024-08-09
> >
> > After reviewing all the materials and discussions on this page, I believe the authors have made significant efforts to address my main concerns. The proposed framework Colabator has strong generalization performance, including dehazing, desmoking and underwater image enhancement tasks. I believe that this quality-evaluation-based pseudo-label selection framework provides a new direction for real-world image restoration.
> > I am now happy to update my rating to strong acceptance.

---

> > > ### Author Response · Authors · 2024-08-09
> > > **Thanks for recognizing the value of our work!**
> > >
> > > We wish to express our sincere appreciation to the reviewer for recognizing the substantial significance of our contribution, specifically the Colabator framework and the CORUN method, within the realm of real-world image dehazing, along with image desmoking and underwater image enhancement. Your acknowledgement holds great importance to us and serves as a meaningful validation of our dedicated efforts to advance this critical area of research.

---

### Official Review · Reviewer_h5fd · 2024-07-10

**Soundness:** 3
**Presentation:** 3
**Contribution:** 3
**Rating:** 7
**Confidence:** 4

**Summary:**

This paper focuses on challenging real-world image dehazing problem. The authors develop their network based on unfolding network, while leveraging cooperative proximal mapping modules to facilitate the estimation of transmission map and image content. In addition, the authors propose an interesting teacher network, named as Colabator, with newly developed quality assessment metrics to better provide the high-quality pseudo-labels for dehazing. The subjective metrics on different benchmark, and the user study with downstream tasks evaluation demonstrate the effectiveness of this work.

**Strengths:**

1.	The Colabator devised by authors sound reasonable, it also provides some insights on training scheme design for restoration problems where paired-images data are hard to collect.
2.	The CPMM modules further improve the representative abilities of unfolding networks to model the gradient descent process of image dehazing problems.
3.	The experimental results give better subjective qualities of the proposed network compared to other methods. Moreover, the results of user study and downstream tasks confirm this point. Judging by myself, I also agree that the authors’ method generates more natural restored images with minimal color shift and the best quality.

**Weaknesses:**

1.	The derivation of equation (7) and (10) are unclear. The authors may need to provide some basic theories before stating these equations.
2.	The details of using DA-CLIP as haze density evaluator is unclear, the statement in line 164 indicates that the authors use a fixed text feature, which is also missing in the article. Moreover, the reason behind the patch partition in line 165 is not clearly stated, the information about patch size N cannot be found in the article.
3.	The article lacks ablation studies about CPMM modules. I wonder how network will performance if removing all CPMM modules.

**Questions:**

First of all, please address my concerns listed in the weakness part. I have some additional questions:

1.	About density loss: similar as the problems in the second point of weakness, how does this loss function formulate?
2.	In the fine-tuning phase, as the student network is optimized using strong augmented data, is the performance gain coming from this augmentation strategy?
3.	Why partition images before quality assessment? How about using full-resolution images as input?

I hope to see the answers and will adjust my rating accordingly.

**Limitations:**

The authors have stated the limitations of their article in the supplementary materials.

---

> ### Author Rebuttal · Authors · 2024-08-06
>
> Thanks for the valuable comments. If not specifically stated, all experiments are conducted on the real-world image dehazing task with *RTTS* for space limitation.
>
> **W1: Unclear derivation of equation**
>
> For space limitation, we only show the detailed derivation of Eq. (7) and the Eq. (10) can be derived similarly.
>
> Having gotten Eqs. (5) and (6), the solution of $\hat{\mathbf{T}}$ can be formulated according to the proximal gradient algorithm:
>
> $$
> \mathcal{T}(\hat{\mathbf{J}}\_{k-1}, \mathbf{T}\_{k-1})=\frac{1}{2}\| \mathbf{P}-\hat{\mathbf{J}}\_{k-1}\cdot \hat{\mathbf{T}}+\hat{\mathbf{T}}-\mathbf{I}\|^2_2+\frac{\lambda_k}{2}\|\hat{\mathbf{T}}-\mathbf{T}\_{k-1}\|^2_2.
> $$
>
> Then we obtain the partial derivative:
>
> $$
> \partial\_{\hat{\mathbf{T}}}\mathcal{T}(\hat{\mathbf{J}}\_{k-1}, \mathbf{T}\_{k-1})=(\mathbf{I}-\hat{\mathbf{J}}\_{k-1})^T(\mathbf{P}-\hat{\mathbf{J}}\_{k-1}\cdot \hat{\mathbf{T}}+\hat{\mathbf{T}}-\mathbf{I})+ \lambda\_k(\hat{\mathbf{T}}-\mathbf{T}\_{k-1}).
> $$
>
> Let the partial derivative be equal to zero, we archieve the closed-form solution for $\hat{\mathbf{T}}$ in Eq. (7).
>
> Similarly, the solution of $\hat{\mathbf{J}}$ can be formulated as
>
> $$
> \mathcal{J}(\hat{\mathbf{T}}\_k, \mathbf{J}\_{k-1})=\frac{1}{2}\| \mathbf{P}-\hat{\mathbf{J}}\cdot \hat{\mathbf{T}}\_k+\hat{\mathbf{T}}\_k-\mathbf{I}\|^2\_2+\frac{\mu\_k}{2}\|\hat{\mathbf{J}}-\mathbf{J}\_{k-1}\|^2\_2.
> $$
>
> The corresponding partial derivative is
>
> $$
> \partial\_{\hat{\mathbf{J}}}\mathcal{J}(\hat{\mathbf{T}}\_k, \mathbf{J}\_{k-1})=-\hat{\mathbf{T}}\_k^T(\mathbf{P}-\hat{\mathbf{J}}\cdot \hat{\mathbf{T}}\_k+\hat{\mathbf{T}}\_k-\mathbf{I})+\mu\_k(\hat{\mathbf{J}}-\mathbf{J}\_{k-1}).
> $$
>
> The closed-form solution for $\hat{\mathbf{J}}$ is presented in Eq. (7) when let the partial derivative be equal to zero.
>
>
> **W2 & Q1 & Q3:  Details of using DA-CLIP and reasons for partition**
>
> Our method for using DA-CLIP as a haze density estimator involves calculating the cosine similarity between the encoded input image and the encoded input text. The specific formula is
>
> $$
> \mathcal{D}(\mathbf{S}^{R}\_{\widetilde{HQ}})=\frac{Enc\_{image}(\mathbf{S}^{R}\_{\widetilde{HQ}})}{\|Enc\_{image}(\mathbf{S}^{R}\_{\widetilde{HQ}})\|}\cdot (\frac{Enc\_{text}(\text{Text})}{\|Enc\_{text}(\text{Text})\|})^\top
> $$
>
> The text we used is "hazy" from the DA-CLIP provided in the text list.
>
> **(Q1)** As Density Loss, we normalize the result of this formula from 0 to 1, the higher result means higher density.
>
> **(W2)** As Haze Density Evaluator for Colabator, the $norm$ in eq. (12) means $1 - normalized(\mathcal{D}(\mathbf{S}^{R}\_{\widetilde{HQ}}))$ which has the same value as (1 - density loss), the higher result means lower density.
>
> **(Q3)** We assess the image both globally and locally. We use global assessment to determine whether the overall quality of the image has improved relative to the images in the existing optimal label pool, updating the pseudo-labels. We also use block-based processing to obtain independent haze density and image quality scores for each region of the image. This allows us to assign appropriate confidence weights to the pseudo-labels of each block. This approach enables our network to maximize the use of high-quality dehazed parts of the image while avoiding being misled by poorly dehazed or low-quality parts. By focusing on both global and local aspects of the image, our method achieves better performance.
>
> ||NIMA↑|BRISQUE↓|FADE↓|
> |---|---|---|---|
> |Only Full|5.229|13.099|0.803|
> |Partition+Full(Ours)|5.315|11.956|0.751|
>
> **(W2)** The patch size N is set to 32 to balance performance and efficiency. We will add this.
>
>
> **W3: Ablation study on CPMM**
>
> After removing all CPMM modules, our method retained only 8 learnable parameters. As shown in the table, removing CPMM  brings a significant performance decline both in the FADE metric, which measures haze removal capability, and the NIMA and BRISQUE metrics, which measure the quality of generated results. However, compared to the original hazy images, the CPMM-removed CORUN still possesses a certain haze removal capability.
>
> ||NIMA↑|BRISQUE↓|FADE↓|
> |---|---|---|---|
> |Hazy(Input)|4.483|36.642|2.484|
> |w/o CPMM|4.836|38.197|1.362|
> |w/ CPMM(Ours CORUN+)|5.315|11.956|0.751|
>
> In the **Figure.A3**, we provide a case of generated result examples. The images include, respectively, the input image, the result with the CPMM module, the result without the CPMM module, and the result without the CPMM module but with manually increased exposure. From our analysis, we can conclude that removing the CPMM module severely affects the flexibility of CORUN, causing the generated results to lose a significant amount of detail, especially in dark areas, and leading to a substantial decrease in haze removal capability.
>
>
> **Q2: Ablation study on strong data augmentation before student network**
>
> We conducted ablation experiments on the data augmentation strategies used during the fine-tuning phase. In the table below, we provide the quantitative results obtained after disabling strong data augmentation for the student network input. From the results, it can be seen that the strong data augmentation strategy for the student network indeed brings some performance gains, but even without strong data augmentation, Colabator still provides a considerable performance improvement to the network. Therefore, we can conclude that the performance gain during the fine-tuning phase does not entirely come from the strong data augmentation strategy.
>
> ||NIMA↑|BRISQUE↓|FADE↓|
> |---|---|---|---|
> |w/o Colabator|4.856|16.541|1.091|
> |w/o Strong aug.|5.084|12.671|0.813|
> |w/ Strong aug. (Ours CORUN+)|5.315|11.956|0.751|

---

> > ### Comment · Reviewer_h5fd · 2024-08-11
> >
> > We thank the authors for providing a detailed rebuttal in such a short time duration. After reviewing the rebuttal, I believe the authors have solved all my concerns about their paper. I will raise my rating to accept, and I support accepting this paper.
> > However, PLEASE make sure to update missing information about DA-CLIP and equation parts in the final revision to make the paper more comprehensive.

---

> > > ### Author Response · Authors · 2024-08-11
> > > **Thanks for recognizing the value of our work!**
> > >
> > > We wish to express our sincere appreciation to the reviewer for recognizing the substantial significance of our contribution, specifically the Colabator framework and the CORUN method, within the realm of real-world image dehazing, along with image desmoking and underwater image enhancement. Your acknowledgment holds great importance to us and serves as a meaningful validation of our dedicated efforts to advance this critical area of research. We will update the missing information about DA-CLIP in our final version.

---

### Official Review · Reviewer_ivCJ · 2024-07-12

**Soundness:** 3
**Presentation:** 3
**Contribution:** 2
**Rating:** 5
**Confidence:** 3

**Summary:**

The paper aims for real-world image dehazing, where the paper tries to handle the difficulties of modeling real haze distributions and the scarcity of real data. For modeling haze, the paper proposes to jointly model atmospheric scattering and image scenes by a cooperative unfolding network. To handle the scarcity of data, the paper proposes to generate pseudo labels using an iterative mean-teacher network, which the paper terms Coherence-based label Generator (Colabator).

**Strengths:**

- The paper is well-written and easy to follow.
- The proposed method is effective in that object detection results are improved on images dehazed by the proposed method in comparison to previous works.
- The iterative mean-teacher framework for generating pseudo labels seems to be effective in image dehazing.

**Weaknesses:**

- The major concern I have for this paper is that the proposed method seems to be mostly combination of existing works.
   - The joint estimation of atmospheric scattering/transmission (or atmospheric light) and clean image is not new:
      - Mondal et al., Image Dehazing by Joint Estimation of Transmittance and Airlight using Bi-Directional Consistency Loss Minimized FCN
      - Im et al., Deep Variational Bayesian Modeling of Haze Degradation Process
      - Zhang et al., Joint Transmission Map Estimation and Dehazing using Deep Networks
      - Ren et al.,  Single image dehazing via multi-scale convolutional neural network
    - Deep Unfolding Networks (DUNs) has been utilized for dehazing by Yang and Sun [50], mentioned in the paper
    - The use of PGD for DUNs in the context of image restoration has been done by DGUN [33], as mentioned in the paper.
    - Also, using mean teacher for creating pseudo labels is commonly performed in deep learning.
   - Considering all of the above, the proposed method seems to be the combination of the existing methods.
   - What is a major difference and technical novelty in comparison to the combination of existing methods above?
- Since the proposed method is a combination of many components, a detailed ablation study would be needed to justify the effectiveness of each component.

**Questions:**

My major concern and questions have been written in the weakness section.

**Limitations:**

The authors have mentioned the limitation of their work in the supplementary document.

---

> ### Author Rebuttal · Authors · 2024-08-06
>
> Thanks for the valuable comments. If not specifically stated, all experiments are conducted on the real-world image dehazing task with RTTS for space limitation.
>
> **W1: Contribution**
>
> Our work goes beyond simply combining existing methods. We introduce the Cooperative Unfolding Network (CORUN), which iteratively optimizes transmission maps and scene information with theoretical guarantees. We also present Colabator, a semi-supervised fine-tuning framework that requires no extra computational cost and adapts well to real-world scenarios with just a few iterations.
>
> Specifically, compared to previous methods, our innovations are as follows:
>
> - Methods by Mondal et al.[64], Im et al.[65], and Ren et al.[66] use common networks to estimate atmospheric light and transmission maps through the atmospheric scattering model but overlook other complex scene information. Zhang et al.[27] estimate the transmission map and haze features from hazy images but ignore the atmospheric model during reconstruction, limiting generalization.
>
>     In contrast, our method mathematically models the relationship between the image scene and transmission map, utilizing a deep unfolding network based on proximal gradient descent for progressive dehazing. This enhances interpretability and detail restoration, resulting in more realistic images. To verify this, we incorporate our joint estimation component into the three open-sourced methods and observe evident performance gain.
>
>     |Datasets|Metrics|Ren[66]|Ren+CORUN|Mondal[64]|Mondal+CORUN|Im[65]|Im+CORUN|CORUN(Ours)|
>     |-|-|-|-|-|-|-|-|-|
>     |O-HAZE|PSNR↑|15.72|19.91|16.37|18.85|18.85|21.36|25.66|
>     ||SSIM↑|0.703|0.753|0.717|0.750|0.776|0.793|0.847|
>     |I-HAZE|PSNR↑|14.49|18.83|15.52|18.27|17.93|20.15|23.90|
>     ||SSIM↑|0.710|0.760|0.721|0.763|0.752|0.785|0.868|
>
> - PDN[50] introduced a deep unfolding network for image dehazing but couldn't effectively use complementary information between the scene and transmission map, leading to sub-optimal results with real-world data. In contrast, our new dual proximal gradient descent cooperative unfolding network considers both atmospheric light and image information, enhancing robustness and flexibility. We integrated our optimization mode into PDN[50], resulting in improved performance.
>
>     |Datasets|Metrics|PDN[50]|PDN+CORUN|DGUN[33]|DGUN+CORUN’s model|DGUN+CORUN|CORUN(Ours)|
>     |-|-|-|-|-|-|-|-|
>     |O-HAZE|PSNR↑|16.37|19.02|18.86|20.16|22.74|25.66|
>     ||SSIM↑|0.717|0.763|0.756|0.788|0.805|0.847|
>     |I-HAZE|PSNR↑|15.86|18.80|18.27|19.82|22.63|23.90|
>     ||SSIM↑|0.712|0.757|0.732|0.763|0.791|0.868|
>
> - DGUN models the relationship between degraded and clean images but ignores the interaction between the scene and transmission map in dehazing. Additionally, DGUN[33]'s unfolding strategy overlooks complex hazy conditions in real-world scenarios. Our CORUN addresses this with CPMM modules and a proposed global coherence loss. Replacing DGUN[33]'s dehazing model with ours and integrating our unfolding framework into DGUN[33] both improve performance.
> - Mean-teacher, commonly used in high-level vision tasks, commonly utilizes uncertainty to constrain pseudo-labels which isn't suitable for low-level vision tasks. Some attempts have been made to adapt it for low-level tasks using data augmentation, but these still face issues with erroneous pseudo-labels and overfitting. Colabator improves this by using CLIP and NR-IQA to weight pseudo-label regions, guiding the student network to focus on high-quality areas. It also evaluates pseudo-label quality globally to refine the optimal label pool iteratively, enhancing performance over Mean-teacher. Comparison results can be found in **Table A10**.
>
> - Totally, the major difference and technical novelty in comparison to the combination of existing methods above is:
>     - From a mathematical optimization perspective, we jointly model the transmission map and image scene based on the atmospheric scattering model, creating a novel optimization function that effectively handles complex real-world haze scenarios.
>     - Our unfolding network, CORUN, is tailored to this optimization formula, integrating scene and haze feature commonalities. Using the proximal gradient descent algorithm, CORUN reconstructs high-quality dehazed images progressively from coarse to fine.
>     - Our Colabator architecture is plug-and-play, designed for low-level vision tasks like real-world image dehazing. It evaluates pseudo-labels globally and locally, helping the student network learn from high-quality regions while minimizing errors. It also dynamically maintains an optimal label pool to ensure pseudo-labels stay globally optimal.
> - Therefore, our work is not a mere assembly of existing methods but an effective framework proposed specifically based on the characteristics and challenges of real-world image dehazing.
>
> **W2: Detailed ablation study**
>
> In the paper, we presented Colabator, along with ablation experiments on the Optimal Label Pool, Trusted Weights, Iterative Mean-Teacher, and the number of layers in the deep unfolding network. **We have supplemented additional ablation experiments in** **Global Rebuttal‘s pdf supplement**, which will be included in the supplementary materials of the final version.
>
> The results of our CORUN and Colabator on various datasets, data processing methods, and types of Image Quality Assessment (IQA) are presented in **Tables A2, A4, A6, A17**. **Tables A3, A14** demonstrate the generalizability of Colabator across different tasks and networks. **Tables A1, A12** detail the ablation studies of CORUN’s components, while **Table A7** shows the ablation of loss functions. **Table A8** assesses the robustness of DA-CLIP. **Tables A9, A11, A13, A15, A16** demonstrate the ablations of Colabator’s components and functions.
>
> [64]Mondal et al., CVPRW, 2018
>
> [65]Im et al., CIKM, 2023
>
> [66]Ren et al., ECCV, 2016
>
> [67]Zhang et al., TCSVT, 2019

---

> ### Comment · Reviewer_ivCJ · 2024-08-13
>
> I appreciate the authors' rebuttal, which has addressed most of my concerns.
> One additional concern I have after reading the rebuttal is the authors have made comparisons against other methods on I-HAZE, O-HAZE, but not on RTTS, a benchmark the authors have used in the main paper. I believe this has to do with the other reviewer's recommendation.
> It would be great if the authors could make comparisons on RTTS as well in the final version.
> Also, the authors have presented a lot of new experimental results during the rebuttal, requiring a major change to the paper.
> I'm ok with the acceptance and hence raise the rating, given that the authors make these major changes to the final version of the paper, along with new comparisons on RTTS as well.

---

> > ### Author Response · Authors · 2024-08-13
> > **Thanks for recognizing the value of our work!**
> >
> > We wish to express our sincere appreciation to the reviewer for recognizing the substantial significance of our contribution, specifically the Colabator framework and the CORUN method, within the realm of real-world image dehazing!
> >
> > First, we will incorporate the experiments and major changes conducted during the rebuttal phase into our revised manuscript. Our initial plan is to include critical experiments, such as detailed comparisons with existing methods and essential ablation studies, directly in the main text, while other experiments will be placed in the supplementary material due to space constraints. To make room for these important experiments, we will move some less critical visual results to the appendix.
> >
> > Additionally, following the approach in **W1**, we have added comparisons with existing methods on the RTTS dataset. As shown in the table, we observed significant performance improvements when integrating either our joint estimation component or our unfolding framework proposed in CORUN. This experiment will also be included in our revised version.
> >
> > Please feel free to let us know if you have any further concerns. We would be happy to address them.
> >
> > | Metrics | Ren[66] | Ren+CORUN | Mondal[64] | Mondal+CORUN | Im[65] | Im+CORUN | PDN[50] | PDN+CORUN | DGUN[33] | DGUN+CORUN’s model | DGUN+CORUN | CORUN (Ours) |
> > | --- | --- | --- | --- | --- | --- | --- | --- | --- | --- | --- | --- | --- |
> > | NIMA↑ | 3.173 | 3.420 | 3.372 | 3.518 | 3.304 | 3.414 | 2.925 | 3.271 | 4.272 | 4.419 | 4.583 | 4.823 |
> > | BRISQUE↓ | 31.335 | 28.441 | 29.951 | 27.773 | 26.906 | 25.103 | 33.186 | 29.342 | 29.753 | 27.728 | 25.308 | 20.944 |
> > | FADE ↓ | 2.065 | 1.878 | 1.732 | 1.626 | 1.719 | 1.633 | 2.159 | 2.006 | 1.663 | 1.535 | 1.376 | 1.051 |

---

> > > ### Comment · Reviewer_ivCJ · 2024-08-14
> > >
> > > Thanks for the update. I have no further concerns at the moment.

---

> > > > ### Author Response · Authors · 2024-08-14
> > > > **Thanks for recognizing our work!**
> > > >
> > > > Your acknowledgment holds great importance to us and serves as a meaningful validation of our dedicated efforts to advance this critical area of research.
> > > >
> > > > In this work, we are the first to propose a plug-and-play iterative mean-teacher framework (Colabator) for real-world image dehazing, along with a robust dehazing algorithm (CORUN) with theoretical guarantees.
> > > >
> > > > It’s important to emphasize that the Colabator framework is not limited to dehazing; it can also be applied to other significant low-level vision tasks, such as image desmoking and underwater image enhancement. In this case, **we believe this work has the potential to inspire advancements across the entire field of low-level vision for real-world image processing.**
> > > >
> > > > We would greatly appreciate any further discussion, as well as a reconsideration of the rating.

---

> > > > > ### Comment · Reviewer_ivCJ · 2024-08-14
> > > > >
> > > > > I would like to note that I have updated the rating, as noted in the previous comment, considering that authors would make the changes to the final version

---

### Official Review · Reviewer_Udmv · 2024-07-13

**Soundness:** 2
**Presentation:** 2
**Contribution:** 2
**Rating:** 5
**Confidence:** 5

**Summary:**

This paper focuses on Real Image dehazing. They propose a cooperative unfolding network (CORUN) to integrate physical knowledge for image dehazing. The proposed CORUN exploits the complementary information between components in Atmospheric Scattering Model. Besides, due to the lack of real paired data, this paper proposes a Coherence-based Label Generator to generate pseudo labels for real hazy inputs. The experimental results demonstrate that the proposed method achieves good dehazing performance.

**Strengths:**

1.	The research direction of this paper is meaningful.
2.	The quantitative results show that the proposed CORUN+ achieves state-of-the-art performance in terms of NRIQA metrics.
3.	The visual results shown in Fig.5 outperform other competing methods.

**Weaknesses:**

1.	There is still room for improvement in the writing of this paper.
a)	Typo in line 167
b)	How to simplify A in line 111 is unclear. According to my understanding, after simplification, P in Eq. 2 has different physical meaning as defined in line 110.
c)	In lin3 172, the same symbols are used to indicate different images.
d)	Line 187 presents two kinds of reconstruction loss, which is inconsistent with the perceptual loss in line 186.
2.	The experiment is a bit unfair. This paper utilizes the data generation pipeline proposed in RIDCP, which has been proved that can improve the real image dehazing performance of dehazing networks. However, the quantitative results of PDN and MBDN reported in Table 1 seem to be calculated by using their original models, rather than re-training on the same training data used in this paper.
3.	The experiment is a bit insufficient.
a)	The author only validates the effectiveness of Colabator on DGUN, neglecting other widely-used dehazing networks.
b)	The full-reference evaluation on real dehazing benchmarks (e.g., O-HAZE, I-HAZE, etc.) can also be an important evidence to support the effectiveness of the proposed method.

**Questions:**

Please see the weakness for more details.
Besides, I was wondering:
1.	What are the essential advantages of CORUN compared to state-of-the-art end-to-end dehazing models? Are there any experiments?
2.	Whether the proposed method can remove medium or dense haze correctly. Since the dehazing results shown in Fig. 4 still have haze residues even if the hazy input is not very dense.
3.	Why use both perceptual loss and contrastive loss simultaneously? There are overlapping parts between them.
4.	How robust is the estimation of haze density by DA-CLIP?

**Limitations:**

The authors have discussed the limitations.

---

> ### Author Rebuttal · Authors · 2024-08-06
>
> Thanks for the valuable comments. If not stated, all experiments are conducted on the RTTS dataset.
> ```
> *Experiment results are placed in the attached PDF for space limitations*
> ```
> **W1: Writing and formula**
>
> - The redundant comma will be removed.
> - The simplified $P$ in eq.(2) differs from the $P$ in line 110. In line 111, the original atmospheric scattering model $P=J\cdot T+A\cdot (I-T)$ is divided by the atmospheric light A, resulting in $P/A=J/A\cdot T+I-T$. We then simplified $P/A$ and $J/A$ to $P$ and $J$, leading to $P=J\cdot T+I-T$. We will reorganize it.
>
> This simplification aims to help integrate atmospheric light $A$, which is closely related to image features, with the scene for better results. Experiments in **Table A1** show that this improves performance by **10.7%**.
>
> - We will correct the notion: pseudo-dehazed image ${\mathbf{P}^{R}\_{\widetilde{HQ}}}\_{i}$ and previous pseudo-label ${\mathbf{P}^{R}\_{Pse}}\_{i}$.
> - In eqs. (15) and (16), we introduced two loss functions for different phases. Eq. (15) includes reconstruction loss and perceptual loss, while eq. (16) substitutes the perceptual loss with a stronger supervised contrastive perceptual loss. We will rename eq. (15) and eq. (16) to Reconstruction-Perceptual Loss and Reconstruction-Contrastive Perceptual Loss.
>
> **W2: Unfair in data augmentation and generation pipeline**
>
> To ensure a fair comparison with RIDCP, we follow its setup by using the same data pipeline (generation + augmentation) and dataset.
>
> To fully show our results, we provide results under two extra settings: 1) We remove the augmentation pipeline, 2) We further replace RIDCP's dataset and generation pipeline with the OTS dataset (commonly used in previous methods).
>
> As shown in **Table A2**, the results verify our method still achieves a leading place under the two settings compared with existing methods.
>
> **W3: Effect of Colabator and results on full-reference**
>
> - We integrate our Colabator with cutting-edge dehazing methods and test them in RTTS. In **Table A3**, we find evident performance gains brought by Colabator: **23.2%** in C2PNet, **22.0%** in FFA-Net, **31.7%** in GDN.
>
> - We further evaluate our methods on the full-reference setting with O-HAZE and I-HAZE. Because we have paired data for training, we drop Colabator and only train our CORUN following the practice of MB-TaylorFormer. **Table A4** shows that our method achieves leading performance with a big margin on both datasets.
>
> **Q1: Essential advantages**
>
> Previous state-of-the-art dehazing methods, such as C2PNet[60], fail to generalize to broader real-world scenarios like the RTTS dataset when trained on synthetic data. RIDCP, with its well-designed data synthesis pipeline, generalizes better to *RTTS*, while is limited by depth estimation errors, causing unrealistic haze residuals.
>
> Our CORUN is specifically designed for real-world scenarios. It integrates deep unfolding networks and proximal gradient descent algorithms to combine physical priors with network learning. Therefore,  as shown in **Table A5**, our method achieves state-of-the-art real-world dehazing results.
>
> **Q2: Haze residues and medium/dense haze removal**
>
> Apart from the problem of the synthetic training data, *i.e.*, the inaccurate depth estimation and the hazy residue from the ground truth, the reasons for failing to completely remove the haze lie in our semi-supervised setting with incomplete supervision. In Figs. 4, it is a common problem that also exists in other comparative methods.
>
> However, as shown in **Fig. A1**, our method has a better dehazing effect compared to existing methods, which is attributed to the joint estimation capacity of CORUN and the generalizability of Colabator.
>
> Besides, we randomly select 300 data in *RTTS* and invite 10 naive observers to score them based on how hazy they are. Then we select the first 100 images as dense hazy data and the 101st-200th images as medium-density hazy data according to the scores. **Table A6** shows our superiority in removing medium-density and dense haze.
>
> **Q3: Loss functions overlap**
>
> The two losses, *i.e.*, eq.15 and eq.16, are employed in different phases with different inputs and their specific parameters in eq.20 and eq.21.
>
> In the pre-training phase (eq.20), we use eq.16 with contrastive learning for constraints, using synthetic data. During the fine-tuning phase (eq.21), we employ both eq.15 and eq.16. Here, eq.15 uses synthetic data to ensure stability in the fine-tuning process, while eq.16, which incorporates contrastive perceptual loss, uses real-world data to provide stronger constraints and align the model training with real-world conditions. As shown in **Table A7**, our ablation verifies that combining these two losses enhances the model's performance. We will reorganize the content.
>
> **Q4: Robustness of DA-CLIP**
>
> We evaluated the robustness of DA-CLIP by randomly selecting 300 images and assessing their density score. We invited 10 observers to rate haze density in these images. In **Table A8**, our analysis revealed that while DA-CLIP's results generally align with human ratings (with discrepancies under 0.2), significant differences occur in certain situations. Notable discrepancies (over 0.5) were observed in cases involving 1) images with predominant sky coverage on cloudy days, 2) color distortion or severe faults, and 3) insufficient lighting at night (examples shown in **Fig. A2**).
>
> To address DA-CLIP's limitations in handling complex scenes, we designed a pseudo-label rating strategy. Along with DA-CLIP, we used NR-IQA to evaluate potential pseudo-labels. A pseudo-label is updated only if its scores from both DA-CLIP and NR-IQA surpass the latest label in our optimal label pool. Locally, we partitioned images and used DA-CLIP and MUSIQ to assign weights, encouraging the network to focus on higher-quality regions with better dehazing effects. **Table A9** verifies the effect of our pseudo-label rating strategy.

---

> > ### Comment · Reviewer_Udmv · 2024-08-13
> > **Thank you for your detailed reponse**
> >
> > The detailed rebuttal from the authors addresses most of my concerns.
> > I want to thank the authors for their work.
> > At this stage, I am considering the overall rating of this work mainly based on its novelty, contributions, and insights.

---

> > > ### Author Response · Authors · 2024-08-13
> > > **Thanks for your encouraging reply!**
> > >
> > > We would like to sincerely thank the reviewer for the encouraging feedback.
> > >
> > > In our **seven** response to reviewer Udmv, we have added **nine** tables and **two** figures to thoroughly demonstrate the superiority of our CORUN dehazing model and the generalizability of our Colabator framework.
> > >
> > > Additionally, we encourage reviewer Udmv to briefly review the content in the global rebuttal as well as our responses to the other reviewers. These sections further validate the generalization of our Colabator framework across other critical low-level tasks and provide a detailed comparison between our method and existing dehazing techniques, supported by extensive experimental evidence.
> > >
> > > We believe this will help the reviewer fully understand the novelty, contributions, and insights of our paper. Specifically, **we are the first to propose a plug-and-play iterative mean-teacher framework (Colabator) for real-world image dehazing, along with a robust dehazing algorithm (CORUN) with theoretical guarantees.**

---

> > > > ### Comment · Reviewer_Udmv · 2024-08-13
> > > > **final score**
> > > >
> > > > I was convinced by the statement from the authors.
> > > > This work is somewhat acceptable.
> > > > I decide to raise my score to 5.

---

> > > > > ### Author Response · Authors · 2024-08-13
> > > > > **Thanks for recognizing the value of our work!**
> > > > >
> > > > > We wish to express our sincere appreciation to the reviewer for recognizing the substantial significance of our contribution, specifically the Colabator framework and the CORUN method, within the realm of real-world image dehazing!
> > > > >
> > > > > Your acknowledgment holds great importance to us and serves as a meaningful validation of our dedicated efforts to advance this critical area of research.

---

### Author Rebuttal · Authors · 2024-08-06

We extend our sincere gratitude to all the reviewers (**R1**-**Udmv**, **R2**-**ivCJ**, **R3**-**h5fd**, and **R4**-**wF18**) for their insightful and considerate reviews, which help us to emphasize the contributions of our approach. We are pleased to hear that the reviewers approved the novelty of our work, as well as the commendable performance (**R1, R2, R3, R4**).

We are delighted to see reviewers confirm our contributions to the field of real-world image dehazing. These encompass our novel deep unfolding dehazing method, CORUN, and the ingenious plug-and-play pseudo-label generation and fine-tuning framework, Colabator.

In direct response to your thoughtful comments, we have methodically addressed each point in our individual responses, and we provide a summary here:

- We corrected writing errors, revised certain formulations, and added information about the theory, formulas, and derivation process in this paper to enhance clarity.
- We compared our method with more SOTA methods on additional datasets to underscore our superiority in performance.
- We added experiments to verify the generalizability of our plug-and-play Colabator framework and the advancement of CORUN.
- We provided more ablation study results to demonstrate the robustness and effectiveness of the various modules in CORUN and Colabator.

Thanks again for all of your valuable suggestions. We will update the paper accordingly and release our code for community study. We appreciate the reviewers' time to check our response and **hope to further discuss with the reviewers whether the concerns have been addressed or not**. If the reviewers still have any unclear parts about our work, please let us know.

```
Due to space limitations, images and most of our supplementary experiments are stored in the attached PDF. Please download this PDF for more information. Thank you!
```

---

### Comment · Area_Chair_97Af · 2024-08-09
**Read the rebuttal and discuss with the authors**

Hi Reviewers,

The authors have submitted their rebuttal responses addressing the concerns and feedback you provided. We kindly request that you review their responses and assess whether the issues you raised have been satisfactorily addressed.

If there are any areas where you believe additional clarification is needed, please do not hesitate to engage in further discussion with the authors. Your insights are invaluable to ensuring a thorough and constructive review process.

Best
AC

---

### Decision · Program_Chairs · 2024-09-25

**Decision:**

Accept (spotlight)

**Comment:**

This paper was reviewed by four experts. The reviewers generally appreciated the central concept and were persuaded by the effectiveness of the proposed technique. Initially, the review scores were varied and included several concerns. However, during the rebuttal phase, the authors effectively addressed most of these issues. As a result, reviewers who had previously given lower scores revised their assessments to favor acceptance. Given the unanimous recommendation for acceptance, AC believes the paper is a clear  acceptance case.